# Deep learning-assisted comparative analysis of animal trajectories with DeepHL

Takuya Maekawa [1✉], Kazuya Ohara[1], Yizhe Zhang[1], Matasaburo Fukutomi[2], Sakiko Matsumoto[3], Kentarou Matsumura[4], Hisashi Shidara [5], Shuhei J. Yamazaki[6], Ryusuke Fujisawa[7], Kaoru Ide[8], Naohisa Nagaya[9], Koji Yamazaki[10], Shinsuke Koike [11], Takahisa Miyatake [4], Koutarou D. Kimura [6,12], Hiroto Ogawa [5], Susumu Takahashi[8] & Ken Yoda[3]

A comparative analysis of animal behavior (e.g., male vs. female groups) has been widely used to elucidate behavior specific to one group since pre-Darwinian times. However, big data generated by new sensing technologies, e.g., GPS, makes it difficult for them to contrast group differences manually. This study introduces DeepHL, a deep learning-assisted platform for the comparative analysis of animal movement data, i.e., trajectories. This software uses a deep neural network based on an attention mechanism to automatically detect segments in trajectories that are characteristic of one group. It then highlights these segments in visualized trajectories, enabling biologists to focus on these segments, and helps them reveal the underlying meaning of the highlighted segments to facilitate formulating new hypotheses. We tested the platform on a variety of trajectories of worms, insects, mice, bears, and seabirds across a scale from millimeters to hundreds of kilometers, revealing new movement features of these animals.

[1] Graduate School of Information Science and Technology, Osaka University, Osaka, Japan. [2] Graduate School of Life Science, Hokkaido University, Hokkaido, Japan. [3] Graduate School of Environmental Studies, Nagoya University, Nagoya, Japan. [4] Graduate School of Environmental and Life Science, Okayama University, Okayama, Japan. [5] Department of Biological Sciences, Hokkaido University, Hokkaido, Japan. [6] Graduate School of Science, Osaka University, Osaka, Japan. [7] Graduate School of Computer Science and Systems Engineering, Kyushu Institute of Technology, Iizuka, Japan. [8] Graduate School of Brain Science, Doshisha University, Kyotanabe, Japan. [9] Department of Intelligent Systems, Kyoto Sangyo University, Kyoto, Japan. [10] Department of Forest Science, Tokyo University of Agriculture, Tokyo, Japan. [11] Graduate School of Agriculture, Tokyo University of Agriculture and Technology, Tokyo, Japan. [12] Graduate School of Science, Nagoya City University, Nagoya, Japan. ✉email: maekawa@ist.osaka-u.ac.jp

Recent advances in sensing technologies such as Global Positioning System (GPS) and computer vision provide "big behavioral data" of animals[1–5]. The challenge is how best to capitalize on such data to understand animal behavior, a challenge that has led to many significant cross-disciplinary research projects combining biology and information science[6–9]. One potentially powerful option involves deep learning artificial intelligence (AI). The recent rapid evolution of this has exceeded the capability of humans in a number of "intelligent" tasks requiring human creativity, including the game of Go[10,11]. Because big behavioral data require substantial effort for experts to analyze manually, and because the complexity of the data threatens to blur the capacity for insight, we believe that deep learning-oriented AI is an extremely promising tool to meet complex data challenges. Deep learning and classic machine learning have been used as support tools to quantify animal behavior (e.g., tracking[1,12] and behavior recognition[13]) to reduce the effort involved in manual data labeling by biologists. In contrast, to take deep learning-assisted research one step further, this study leverages deep learning to assist high-level intelligent tasks associated with researchers requiring their insight, for example, the proposal of a hypothesis. Here, we showcase an example of this type of deep learning-assisted research, presenting a computational method that supports the comparative analysis of big behavioral data acquired by, and for, biologists.

Comparative methods have been used by biologists since pre-Darwinian times. Today, with the advent of animal-tracking technologies, comparative analysis, that is, comparison between two groups, for example, experimental vs. control groups and male vs. female groups, is one of the most fundamental approaches to animal behavior analysis. Regarding this, biologists have applied classic knowledge-driven approaches thus far, which are illustrated in the upper portion of Fig. 1a. In this approach, the biologists typically visually compare huge amounts of time-series movement data, such as hundreds to thousands of trajectories, to identify the behavior that characterizes one group for elucidation, for example, sex-specific movement strategies, which requires substantial effort from the researchers. Then, based on the finding, the biologists design some statistical value computed from the behavioral data that well separates the two groups, which is called as a high-level feature in this study. After that, the biologists validate the finding using the computed high-level features with a statistical test (e.g., testing the significant difference in the high-level features between the male and female groups).

However, this approach possesses the potential risk of researchers overlooking an important high-level feature. This problem is obvious in the big data era. Although trajectory analysis based on classic machine learning has been studied, it still relies on features handcrafted by experienced researchers based on findings discovered by manually browsing a large amount of behavioral data or high-level features designed based on hypotheses formulated in advance, yielding a narrowly focused analysis.

In this study, we present a data-driven approach based on deep learning to support an analysis by biologists, as illustrated in the lower part of Fig. 1a. Specifically, this study focuses on a comparative analysis, and a deep learning-based method is proposed to help identify the differences between the trajectory data of two groups. With this approach, to extract the high-level features from the trajectory data for a classification of the two groups, we leverage the feature learning capacity of deep learning, that is, learning of the high-level feature extraction processes performed within a deep neural network (DNN), which was originally conducted by experienced researchers. Although a DNN can extract high-level features objectively, unlike a classic approach, a DNN is regarded as a black box, making it difficult to interpret

the meaning of the high-level features learned by the DNN, that is, to observe the group differences detected by the network. To address this problem, we developed DeepHL, a free, user-friendly, web-based software, in which an interpretable neural network with multi-scale layer-wise attention[14] is used to elucidate the characteristic segments in the trajectories to which the proposed DNN model focuses on in order to distinguish between the trajectories of the two groups (Fig. 1b, c). Because this method informs researchers regarding "which parts of the trajectories they should focus on for further analysis," researchers can save time and effort related to an otherwise manual analysis of huge numbers of trajectories to derive the characteristic segments. In addition, DeepHL finds handcrafted features prepared in advance that are highly correlated with the identified segments to help the researchers consider how best the segments can be explained. Thus, this method facilitates data-driven research for a comparative analysis by supporting knowledge discovery from the data. Figure 1b shows example outputs of DeepHL when we compare trajectories from male seabirds to those from female seabirds. DeepHL automatically finds trajectory segments characteristic of each sex and then provides visualized trajectories highlighting the relevant segments to researchers. Based on the highlighted trajectories and highly correlated features, biologists develop a new hypothesis related to sex-related difference. Then, the biologists can design a high-level feature to validate the hypothesis.

In this study, we present our analysis on a variety of movement trajectories of worms, insects, and mice in laboratories, and animals in the wild, such as bears and seabirds using DeepHL. Behavioral data of these animals were provided by specialists who have been intensively studying the behavior of these animals by manual analysis and/or classic machine learning. We showed the ability of DeepHL to discover new biological insights that have not been found by the manual analysis or classic machine learning. We believe that DeepHL, a web-based open system, could be the first step to democratize AI for biologists who would otherwise have difficulty setting up computing environments for deep learning.

## Results

Here, we briefly introduce the pipeline of the proposed method: (i) DeepHL first trains our proposed network (hereafter called DeepHL-Net) on the trajectory data from two classes. (ii) The attention mechanism in DeepHL-Net then calculates the attention value of each data point in each trajectory for each layer in DeepHL-Net. (iii) Once the attention values are computed, some parts of the trajectories are highlighted by DeepHL using the attention output from a particular layer that is assumed to capture differences in the two classes. To help a user of DeepHL find such a layer (hereinafter, a "discriminator layer") in DeepHL-Net, DeepHL calculates the score for each layer based on attention outputs from the layer. (iv) DeepHL also supports the user in explaining the meaning of the highlighted segments based on a list of handcrafted features from the trajectories prepared in advance by calculating the correlation between the attention values and each of the handcrafted features (Fig. 1d).

Before describing our method in detail, herein we provide definitions of the features used in this study. Primitive features are basic features widely used in a locomotion analysis, that is, speed and relative angular speed. Handcrafted features are low-level features handcrafted by researchers, such as acceleration and distance from the initial position, and include the primitive features. High-level features are designed by researchers and characterize a group through a comparative analysis. A high-level feature is computed from the entire trajectory, such as the average

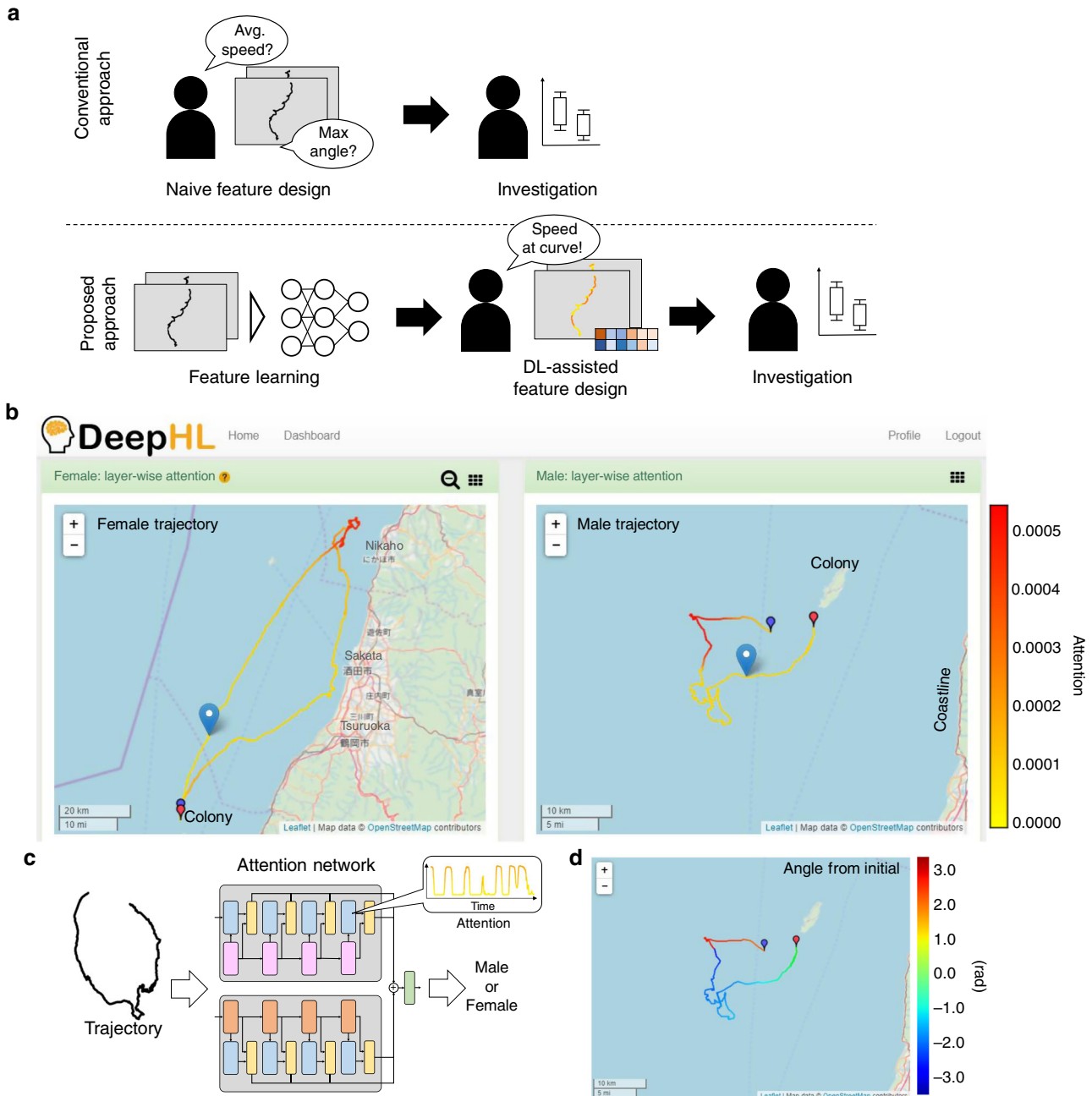

**Fig. 1 Trajectory highlighting and analysis with DeepHL. a** Difference in research procedures between the conventional and proposed deep learning (DL)-assisted approaches. **b** Screenshot of the DeepHL web interface comparing the trajectories of a female (left) and male (right) streaked shearwater. Some characteristic segments of the trajectories are highlighted in red; these were detected by our neural network model trained to distinguish between the trajectories of male and female birds. A user can observe that there is something worth investigating in the highlighted segments. In this case, the female trajectory is highlighted when the female bird stays close to the coastline (see Supplementary Information, Application to the study of seabirds, for more detailed analysis). In contrast, the male trajectory is highlighted when the male bird travels away from the coastline. Note that the small blue and red pins on the maps indicate the starting and terminating points of the trajectories, respectively. The large blue pin on the map moves along the trajectory at a speed proportional to the actual movement speed (Supplementary Movie 1). **c** DeepHL extracts the segments in an input trajectory to which the neural network pays attention when classifying the trajectory by using a time series of the attention values. These segments reveal the importance of each data point. The input trajectory is colored by the time series of the attention values. The range of colors used to color the trajectories is shown on the right of **b**. **d** To facilitate a deeper understanding of the implications of the highlighted segments, DeepHL colors trajectories with the values of other sensor data or handcrafted features highly correlated with the attention values; the angle between the vertical axis (y axis) and a line segment connecting the initial position and each point is used in this example. Points with large angles are focused as shown in the red segments in the male bird trajectory of **b**. Base map and data copyright OpenStreetMap contributors (License: www.openstreetmap.org/copyright).

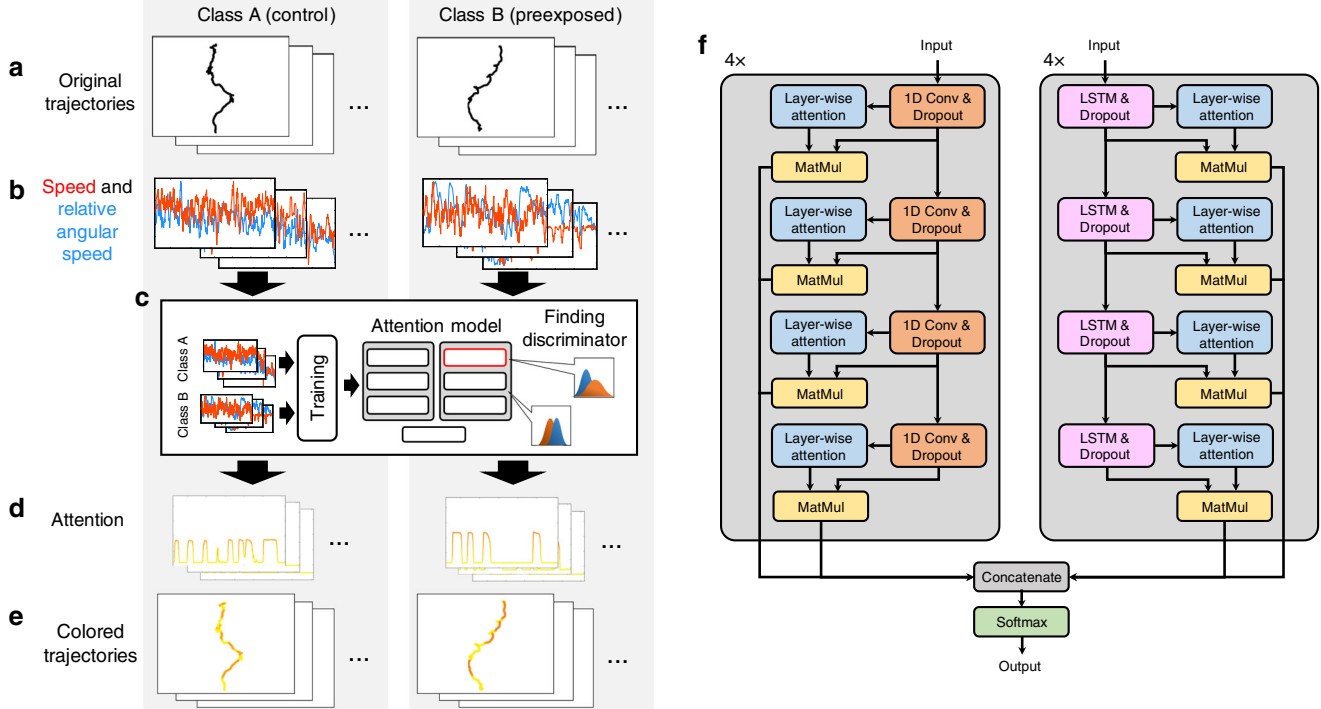

**Fig. 2 DeepHL trajectory highlighting.** We assume that the trajectories of two classes are given: class A and class B in this example, which corresponds to worms without and with prior odor learning, respectively. **a, b** Trajectories, that is, a time series of two-dimensional coordinates, are converted into time series of speed and relative angular speed to achieve position- and rotation-invariant analysis. **c** DeepHL-Net is trained on the time series and then a discriminator layer is found using its attention values. **d** The discriminator layer outputs a time series of attention values when a trajectory is fed into the trained DeepHL-Net. The length of the time series is identical to that of the time series of speed and relative angular speed. **e** Each trajectory is colored with its corresponding attention values obtained by the layer. In our system, a large attention value is encoded as red and a small attention value is encoded as yellow, as shown in Fig. 1b. **f** Proposed multi-scale layer-wise attention model (DeepHL-Net). The input and output of this model are the time-series primitive features and predicted class, respectively. The model consists of four stacks for the convolutional layers and four stacks for the LSTM layers to extract features at different levels of scale. Blocks labeled "1D Conv and Dropout" and "LSTM and Dropout" indicate the 1D convolutional layer and long short-term memory (LSTM) layer with dropout, respectively. The "Layer-wise attention" block calculates the attention of the outputs of a convolutional/LSTM layer using Eq. (1). The "MatMul" block multiplies the attention and the outputs of the layer to reflect the segments paid a high level of attention in the classification result. The "Softmax" block indicates the output softmax layer. For more details about the model, see "Methods" section.

movement speed and duration of stay at a feeding location. Although a DNN can also acquire high-level features or concepts, we found it difficult to comprehend these high-level features. A feature calculated in each layer in the DNN is simply called a feature. We explain our method in detail as follows.

**Trajectory highlighting in DeepHL**. We first explain the method of highlighting trajectories using deep learning. Our method assumes that there are two groups of animals with different properties, that is, class A and class B, and each trajectory belongs either to class A or B. We first convert the time series of the coordinates into time series of movement speed and relative angular speed (Fig. 2a, b), which are widely used primitive features indicating movement velocity and orientation[15–17], to achieve position and rotation-invariant trajectory analysis. (For animals that freely move on an agar plate, for example, their absolute coordinates are meaningless.) These features are then fed into DeepHL-Net. DeepHL allows a user to easily input other time series into DeepHL-Net, for example, original coordinates, other primitive features, and other sensor data.

DeepHL-Net is designed to classify a trajectory into either class A or B. We train DeepHL-Net on the extracted time series associated with their class labels (Fig. 2c). Within DeepHL-Net, we identify a discriminator layer that detects characteristic

segments, which is detailed later. Because DeepHL-Net is also designed to output the segments in a trajectory to which the discriminator layer pays attention, we color the trajectory using the attention information (Fig. 2d, e). Figure 2f shows the architecture of the proposed multi-scale layer-wise attention model (DeepHL-Net) comprising eight stacks of 1D convolutional or long short-term memory (LSTM) layers. Because different filter sizes are used in different convolutional stacks, these stacks are designed to extract features at different levels of scale. In addition, the 1D convolutional layers (orange-colored blocks in Fig. 2f) extract short-term features. In contrast, the LSTM layers (pink-colored blocks in Fig. 2f) tend to extract features reflecting long-term dependencies. Furthermore, more abstract features tend to be extracted in deeper layers in each stack. Therefore, the model is designed so that the layers extract features at different levels of temporal scale to classify trajectories. To elucidate which segments of the trajectories are considered to be important by each layer, we introduce an attention mechanism[14] into the model. As shown in Fig. 2f, the outputs of each 1D convolutional/LSTM layer for an input trajectory are used to compute attention as follows:

$$\mathbf{a} = \text{softmax}\left(\tanh(W_a Z^T + b_a)\right). \tag{1}$$

Here, $\mathbf{a} \in \mathbb{R}^{1 \times l_{\text{MAX}}}$, which shows the importance (i.e., attention) of each data point in the trajectory and is also used to color

the trajectory, is an attention vector that has the same length as the trajectory, where $l_{MAX}$ is the maximum length of the input trajectories. Matrix $Z \in \mathbb{R}^{l_{MAX} \times N}$ is an output matrix of the 1D convolutional/LSTM layer, where $N$ is the number of nodes in the convolutional/LSTM layer. Finally, $W_a \in \mathbb{R}^{1 \times N}$ and $b_a \in \mathbb{R}^{1 \times l_{MAX}}$ are the weight matrix and bias, respectively. The softmax function ensures all the output values sum to 1, and the tanh function limits the output value of its input to a value between $-1$ and 1. Equation (1) is implemented as an artificial neuron in DeepHL-Net ("layer-wise attention"; aqua-colored blocks in Fig. 2f). The attention is multiplied by the outputs of the 1D convolutional/LSTM layer to contrast the segments to which the layer pays attention ("MatMul"; khaki-colored blocks in Fig. 2f). The multiplied outputs of all layers are concatenated and then used to output an estimate, that is, class A or B, in a densely connected output layer using the softmax function, that is, the final layer in DeepHL-Net (green-colored block in Fig. 2f). As mentioned above, our model is designed to calculate attention information at different levels of scale (see "Methods" section for more details about the model).

**Comparative analysis using DeepHL.** A user of DeepHL discovers knowledge using a web page that displays highlighted trajectories (Fig. 1b). We explain the usage of DeepHL through an analysis of the roundworm *Caenorhabditis elegans*, which is commonly used as a model animal in neuroscience to understand how learning modulates behavior[18,19]. Previous studies revealed that worms learn prior experience of the repulsive odor 2-nonanone in dopamine-dependent manner[20,21]: the worms preexposed to the odor migrate further away from the odor source more efficiently than naive worms do. Interestingly, the average speeds of the worms with or without odor learning are not significantly different, suggesting that the preexposed worms avoid the odor more efficiently. To comprehensively determine the high-level behavioral features characteristic of the repulsive odor learning, we compared the trajectories of naive worms (control class; 163 trajectories) to those of worms preexposed to the odor (preexposed class; 162 trajectories) using DeepHL. The positions of each worm's centroid on a 9-cm agar plate were recorded at 1 Hz for 600 s (Fig. 3a; see "Methods" and Supplementary Table 2). DeepHL-Net was trained on a multivariate time series of primitive features that DeepHL automatically extracts from the time series of trajectories (see "Methods," Supplementary Information, Algorithm, and Supplementary Table 1). Here, the classification accuracy of the trained DeepHL-Net was 93.9% (see "Methods"), indicating that DeepHL-Net was properly trained. When the accuracy is low, for example, 50%, we can regard such a state as having no differences between the two classes or the training data having certain problems (e.g., an excessively small amount of data).

In the following, we explain the process of knowledge discovery using the functions of DeepHL:

1. Screening layers: Because DeepHL-Net comprises several layers, DeepHL helps the user find discriminator layers by computing a score for each layer using the following criteria:

- A discriminator layer should pay attention only to a portion of a trajectory. Technically speaking, an attention vector from the discriminator layer should have large values within limited segments. When the attention values are identical throughout the entire trajectory, the user cannot determine which part of the trajectory is characteristic of the class of interest.
- It is desirable that the way attention is paid to the segments of trajectories belonging to one class by the layer is different

from that for another class. Technically speaking, a distribution of attention values using the layer for one class should be different from that for another class. For example, when the layer exhibits large attention values to segments in trajectories belonging to only one class, the user can easily understand that these segments are characteristic of that class.

See "Methods" section for an equation to calculate the score. The DeepHL web interface provides a ranking of the layers based on the calculated scores, enabling the user to easily find high-scoring layers, which can provide an insightful highlight of the trajectory.

2. Showing colored trajectories: In this stage, the user compares trajectories colored by the identified discriminator layer. In the example of Fig. 3b (colored by a discriminator layer with the highest score), only the relatively straight segments of the trajectories are highlighted in red. In contrast, the layer does not pay attention to segments representing more complex movement (yellow segments). The straight and complex movements reflect the two major behavioral states of the worms: "run" and "pirouette"[15,19]. DeepHL found that the run behavior of the preexposed class differs from that of the control class.

Note that, because the number of trajectories to be analyzed is large in many cases, DeepHL has a function for screening the trajectories when the user attempts to show highlighted trajectories by a discriminator layer. Especially when we deal with the trajectories of wild animals, not all trajectories include segments characteristic of a specific class. Therefore, DeepHL computes a score of each trajectory, enabling the user to focus mainly on, for example, trajectories with "female-like" segments. The score is calculated as $V(\mathbf{a})$, where $\mathbf{a}$ is a time series of attention of the trajectory, to find trajectories with characteristic segments. When the variance value is small, this indicates that the layer does not pay attention to particular segments in the trajectory. In addition, the DeepHL web interface provides a classification result for each trajectory, permitting the user to ignore misclassified trajectories when the user browses trajectories.

3. Understanding meaning of highlights: DeepHL provides two functions to help the user understand the reason why a segment attracts attention using a discriminator layer. The first function provides the correlation between the time series of attention values and each of computed handcrafted features prepared in advance (or other sensor data). This reveals which handcrafted feature is related to the attention of the layer (Supplementary Table 1). The second function provides the difference in distributions of each handcrafted feature among the two classes within highlighted segments. This reveals which handcrafted feature has different distributions among the two classes within highlighted segments (Supplementary Information, Algorithm).

Note that these handcrafted features are intended to help interpret the meaning of the attention of DeepHL-Net and that the handcrafted features do not always completely explain the meaning of the attention. As shown in the animal studies below, the biologists understand the meaning of such attention and then manually design interpretable high-level features, which are used in statistical tests, with the help of the functions.

In the worm example, the absolute correlation coefficient between the attention values of the layer and the moving average of the worm speed is the highest among all handcrafted features (Supplementary Table 3). Therefore, we then focus on the speed of the worms. Figure 3c shows the moving averages of speed associated with the attention values. Here, we can employ the second function to reveal the difference in speed between preexposed and control worms within highlighted segments. Interestingly, the difference in distributions of speed itself

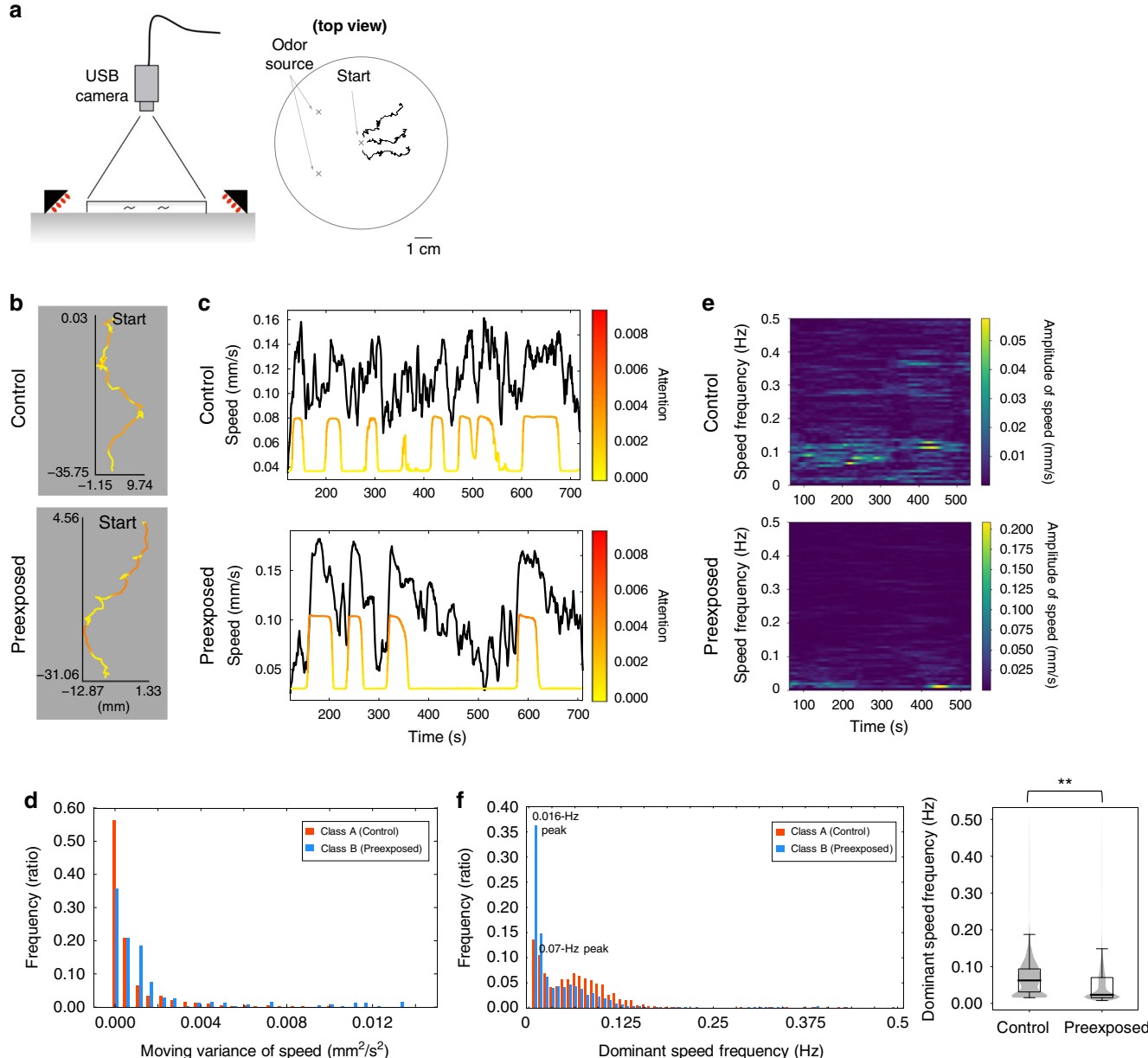

**Fig. 3 DeepHL analysis of repulsive odor learning in worms. a** The experimental setup (left) for monitoring the worm's trajectory (right). **b** Example trajectories of worms colored by attentions of a discriminator layer. Segments of the trajectories corresponding to the run state of the worm are highlighted (in red). **c** Time series of the moving average of speed (black lines) associated with attention values (colored lines). The upper and lower graphs are obtained from the upper and lower trajectories shown in **b**, respectively. **d** Histograms showing the distributions of the moving variance of speed for each time slice within the highlighted trajectory segments. **e** Frequency analysis of the velocity of a preexposed or control worm. A 128-s-wide sliding window was shifted in 1-sample intervals and the amplitude of each frequency component was obtained from its fast Fourier transform (FFT). The upper and lower spectrograms were, respectively, obtained from the upper and lower trajectories shown in **b**. **f** Frequency analysis of the velocity of all the preexposed or control worms computed from entire trajectories. The histograms and box plot show the distributions of the dominant frequency of speed for each time slice. The dominant frequency is the one with the largest amplitude within each window. Significant difference in the dominant frequencies were observed by a generalized linear mixed model (GLMM) with Gaussian distributions ($t = -6.60$; d.f. = 322.8; $p = 1.68 \times 10^{-10}$, effect size($r^2$) = 0.232; **$p < 0.01$; see "Methods"). The $p$ value is two sided. The box plot shows the 25–75% quartile, with embedded bar representing the median; lower whiskers show Q1 $- 1.5 \times$ IQR (Q1: 25% quartile; IQR: interquartile range); upper whiskers show Q3 $+ 1.5 \times$ IQR (Q3: 75% quartile). Control: $n = 76, 784$, preexposed: $n = 75, 750$.

between preexposed worms and control worms within highlighted segments is smaller than that of the moving variance of speed (0.22 vs. 0.25; see Supplementary Information, Algorithm for detailed description about difference computation). DeepHL also provides a graph of the distributions as shown in Fig. 3d. The graph indicates that the changes in speed of preexposed worms

are larger than those of control worms. As shown in the graph related to a control worm (Fig. 3c, upper panel), we can see that, when attention values are high (colored line), the speed indicates tiny high-frequency changes (black line). In contrast, in one typical example of a preexposed worm (Fig. 3c, lower panel), when attention values are high, the worm accelerates substantially

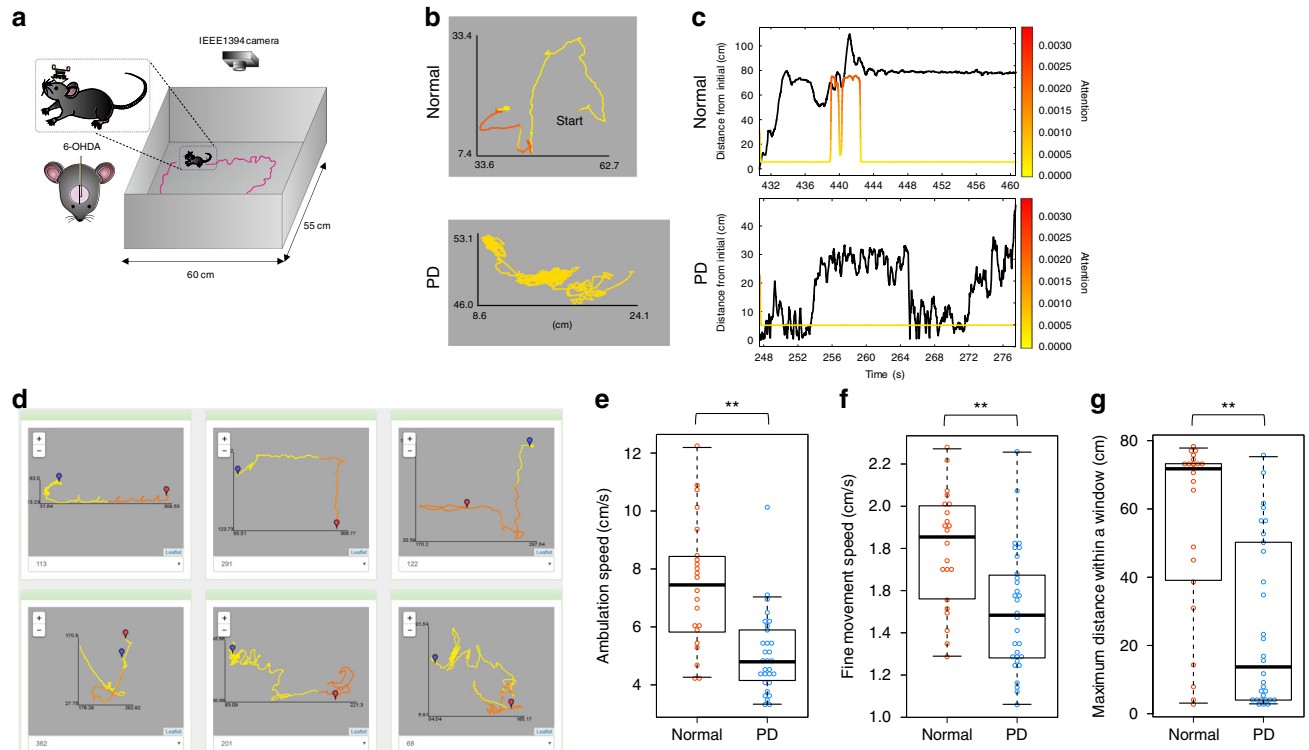

**Fig. 4 DeepHL analysis of normal and PD mouse behavior. a** Experimental apparatus and lesion protocol. **b** Example trajectories of mice colored by attention of the discriminator layer. The upper one is a trajectory of a normal mouse and the lower one is a trajectory of a 6-hydroxydopamine (OHDA) lesion mouse model of Parkinson's disease (PD). The upper trajectory shows that when the mouse is far away from the initial position, the layer pays attention to the corresponding segments (red segments). **c** A time series of the straight-line distance from the initial position (black lines) associated with attention values (colored lines). **d** A screenshot of DeepHL for comparing multiple trajectories colored by the discriminator layer at a glance (normal mice). A blue balloon shows the initial position of each trajectory. **e-g** Average movement speed during ambulation periods, average movement speed during fine movement periods, and average maximum distance within a ±60-s window in a session of normal and PD mice for each entire trajectory (see "Methods"). Significant differences between normal and PD mice were observed for all three features (Wilcoxon rank-sum test, $p = 3.486 \times 10^{-5}$, $p = 5.869 \times 10^{-4}$, $p = 2.666 \times 10^{-4}$; **$p < 0.01$; $n = 22$ original 10-min trajectories from normal; $n = 30$ original 10-min trajectories from PD). The $p$ values are two sided. The edges of the box plot correspond to 95% confidence intervals, the embedded bar represents the median, and whiskers show minimum and maximum values. Dots show values for individual sessions.

and maintains a high speed, resulting in large low-frequency changes in speed as well as large moving variance of speed. DeepHL seems to detect the low-frequency changes in the speed of the preexposed worms as a characteristic behavior of the preexposed worms. Consistently, the lower frequency components of the speed of the preexposed worms are more dominant than those of the control worms (Fig. 3e, f). These results suggest that, in worm odor avoidance behavior, two states for periodic changes in velocity—long-term changes with a peak at 2/128 Hz (i.e., 0.016 Hz; 64 s cycle) and short-term change that peak at 9/128 Hz (i.e., 0.07 Hz; 14.2 s cycle)—exist (Fig. 3f), and that learning modulates the ratio between these two states to avoid odors efficiently. It is reasonable to speculate that maintaining a high speed (resulting in long-term speed changes) only during the run state contributes to efficient odor–source avoidance behavior. Note that these results were not predicted before this analysis because the average velocities of preexposed and control worms are essentially similar[20,22]. The biological significance of the worm and other animal analyses are described in the Supplementary Information.

**Application to the study of mice.** To test general applicability of DeepHL, we compared the behavioral patterns of normal and Parkinson's disease (PD) mice freely moving in an open field (Fig. 4a). Although the primary cause of PD is considered to be the loss of dopaminergic inputs to the striatum, the type of motor symptoms it induces remains unclear. Neurotoxic lesion animal models of PD have been utilized to elucidate the neuronal mechanisms underlying PD. However, in such models, the degree of dopaminergic cell loss can only be established post mortem. To estimate the degree of cell loss before death, several behavioral tests have been developed[23,24]. For instance, frequencies of ambulation, immobility, or fine movement epochs in open-field tests are evaluated. We compared normal mice to PD mice using DeepHL to discover a new high-level behavior feature. The classification accuracy for the mouse dataset is 74.7% (see "Methods" for further details). Figure 4b shows typical examples of trajectories highlighted using a discriminator layer. Segments of the normal mouse trajectory that are far away from the initial position are highlighted in red (see also Fig. 4d). In addition, DeepHL indicated that the attention values highly correlate with the straight-line distances from the initial position (highest; Supplementary Table 3). As shown in Fig. 4c, when straight-line distances from the initial position exhibit high values, the attention values also increase. This result indicates that the behavior of visiting locations far away from the initial position is characteristic of normal mice.

To investigate the usefulness of this finding in terms of PD mouse detection, we designed a new high-level movement feature based on it: the maximum straight-line distance within a ±60 s window. We compare its performance to the performances of existing high-level movement features, that is, ambulation and fine movement speeds (Fig. 4e–g). We compute these three feature values for each entire trajectory and then evaluated the features using information gain[25], which is used to evaluate classification features. A larger value of information gain indicates better classification performance. While the ambulation speed, fine movement speed, and maximum distance all exhibited statistical differences between the normal and PD groups, their information gains were 0.269, 0.184, and 0.287, respectively, indicating that the maximum distance is more useful for evaluating PD symptoms than conventional measures.

The results suggest that normal mice prefer exploring unvisited locations. This feature strongly relates to the straight-line distance from the initial position and differs from widely used existing high-level movement features based on speed. It is well known that rodents such as mice and rats spontaneously prefer to explore an environment, particularly in novel places. Thus, DeepHL may have revealed that the abnormal behavior of PD mice hinders such spontaneous behavioral traits.

**Application to the study of red flour beetles**. In addition to the PD and normal mice, DeepHL was used to detect dopamine-dependent differences in the trajectories of insects. Tonic immobility (TI), sometime called as "thanatosis" or "death-feigning," is an antipredator behavior of many animals[26,27]. Miyatake et al.[28] performed a two-way artificial selection for the duration of TI, and established the strains with short (S strain) and long (L strain) duration of TI in the red flour beetle, Tribolium castaneum. Tribolium castaneum is an insect model species for which all the genomes are already known[29]. The S strain showed significantly higher levels of brain dopamine expression and a higher locomotor activity than those of the L strain[30]. In the present study, we analyzed 419 walking trails collected from S- and L-strain beetles on a treadmill using DeepHL (Fig. 5a and Supplementary Table 2).

The classification accuracy for the beetle dataset is 84.5% (see "Methods" for further details). Figure 5b shows typical examples of trajectories highlighted using a discriminator layer. The trajectories in Fig. 5b appear to be highlighted when the beetles turn, which is the characteristic difference between the L and S strains detected by DeepHL. Consistently, the difference in distributions of the angle from the initial position between the S and L strains within highlighted segments is large (0.61). We can clearly see that the turn in the S-strain trajectories is sharp, and we found similar patterns in other trajectories. (See Fig. 5d, generated by a function of DeepHL that allows the comparison of multiple trajectories at a glance.) Figure 5c shows the angle from the initial position and attention values used for highlighting trajectories in Fig. 5b, indicating increases in attention values just before increases in the angle for the S strain.

As shown in Fig. 5e, we computed an angle of a trajectory segment for each point. Figure 5f shows the distributions of the angles for the S and L strains (the number of data points for the S strain is 185,884 and the number of data points for the L strain is 219,497). We found that the angle for the S strain is significantly smaller than that for the L strain, indicating that the S-strain beetles walk with more angle changing. This finding related to angle change, which has not been discovered by prior studies[30], may lead to new hypotheses concerning the survival strategy of the beetles. The L-strain beetles are known to perform death-feigning as an antipredator behavior. In contrast, the S-strain beetles are assumed to select a survival strategy of changing movement directions to escape from predators.

**Application to the studies of crickets and animals in the wild**. We also employed DeepHL to analyze context-dependent modulation of escape behavior in field crickets, Gryllus bimaculatus. Fukutomi et al.[31,32] revealed that an acoustic stimulus at high frequency (>10 kHz) preceding an air puff alters crickets' moving direction in wind-elicited escape behavior, suggesting that the crickets recognize the high-frequency sound as the echolocation signal of bats and change their behaviors in the presence of predators. Here, we adopted DeepHL to compare two groups of escape movement: prestimulated and control (no sound). In this analysis, in addition to the speed and relative angular speed, we input additional sensor data measured using a treadmill, that is, a rotational speed of the body-axis computed from a body-axis angle measured using the treadmill, into DeepHL-Net. Figure 6a shows typical trajectories colored by the attention values of a discriminator layer. DeepHL shows that the rotational speed of the body axis transiently elevated and peaked earlier in the prestimulated group (Supplementary Information, Application to the study of crickets; Fig. 6b), indicating that the sound preceding the air puff provoked the prompt rotational changes of the body axis.

In addition, we applied DeepHL to the trajectories of wild animals. Figure 6c shows GPS trajectories of female and male seabirds highlighted by a discriminator layer that pays attention to the migration direction of the birds from their colony. Our investigation revealed that the GPS measurements of the female seabirds are significantly closer to the coastline than those of the male seabirds. In this analysis, in addition to the speed and relative angular speed, we input the absolute coordinates (longitude and latitude) into DeepHL-Net because the absolute coordinates of the specific places such as colonies and feeding sites can affect the behavior of the seabirds. The longitude values were highly correlated with the attention values for the female seabirds. Because the coastline runs north–south, the distance between the coastline and a position is related to the longitude of the position. Therefore, this fact indicates that the behavior of the female seabirds is strongly affected by the distance from the coastline (see Supplementary Information, Application to the study of seabirds). As described above, because we can input an additional time series in addition to the speed and relative angular speed into DeepHL, we can see the effect of the time series on the animal behaviors. Figure 6d shows the trajectories of female and male bears highlighted by a discriminator layer that pays attention to male trajectories when a male bear travels a long distance after/before it remains in one place. Our investigation revealed that the male bears combined long distance movements with short rests at many locations and the female bears remained in limited locations for a long time (see Supplementary Information, Application to the study of bears).

In the above analysis of the worms, mice, insects, seabirds, and bears, we could discover findings that were not revealed through a manual analysis or classic machine learning. Here, we can easily observe the differences between two groups from the highlighted trajectories of the seabirds and bears. Specifically, it is impossible to observe the relationships between the preferred locations of female seabirds and coastlines without visualization. For the mouse and beetle studies, we also observe the differences between the two groups from the highlighted trajectories. In contrast, it is difficult to find any differences between the two groups related to the worms and crickets by just browsing the highlighted trajectories. Therefore, leveraging both visualization functions

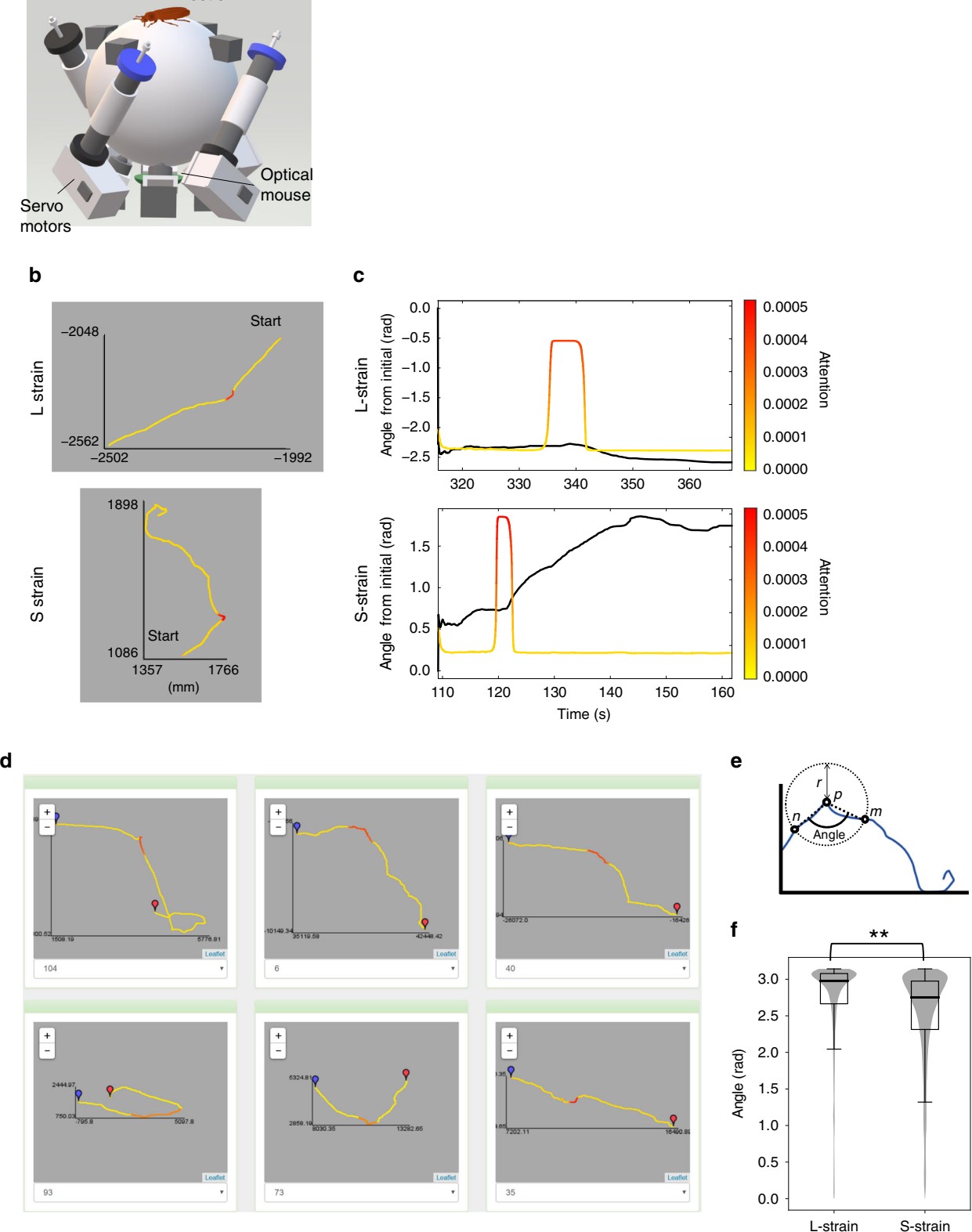

and functions to help interpret the meanings of highlights is important to discover knowledge, which is facilitated by the DeepHL web interface (see also Supplementary Information, User guide to DeepHL).

## Discussion

In this study, we demonstrated that DeepHL is able to extract group differences in trajectories for a variety of taxa that operate across scales (for a quantitative evaluation of DeepHL using

**Fig. 5 DeepHL analysis of the red flour beetles. a** Experimental apparatus (treadmill). **b** Example trajectories of the red flour beetles colored by the attention values of a discriminator layer. The upper one is a trajectory of the L-strain (long-strain) beetle and the lower one is a trajectory of the S-strain (short-strain) beetle. These trajectories show that segments corresponding to orientation change are highlighted. **c** Time series of the angle from the initial position (black lines) associated with attention values (colored lines). The upper and lower graphs are obtained from the upper and lower trajectories shown in **b**, respectively. The lower graph shows that the attention values have large positive values just before the angle increases. **d** Other trajectories belonging to the S-strain class colored by attention of the discriminator layer. **e** We assume a circle centered at each point ($p$) on a trajectory with radius $r$ (100 mm) and obtain points $n$ and $m$ where the trajectory first crosses the circle before/after $p$. We then compute the angle between a line segment connecting $p$ and $n$ and one connecting $p$ and $m$, showing the curvature around $p$. **f** Angles of the L and S strains. The box plot shows the 25–75% quartile, with embedded bar representing the median; lower whiskers show Q1 − 1.5 × IQR (Q1: 25% quartile; IQR: interquartile range); upper whiskers show the maximum values, that is, $\pi$, with the violin plots showing the distributions of data points. Significant difference between the L and S strains was observed using the two-sided ANOVA ($F = 12.57$; d.f. = 1; $p = 0.001$; effect size ($\eta^2$) = 0.09; **$p < 0.01$; see "Methods"). # of data points for S strain is $n = 185, 884$; # of data points for L strain is $n = 219, 497$.

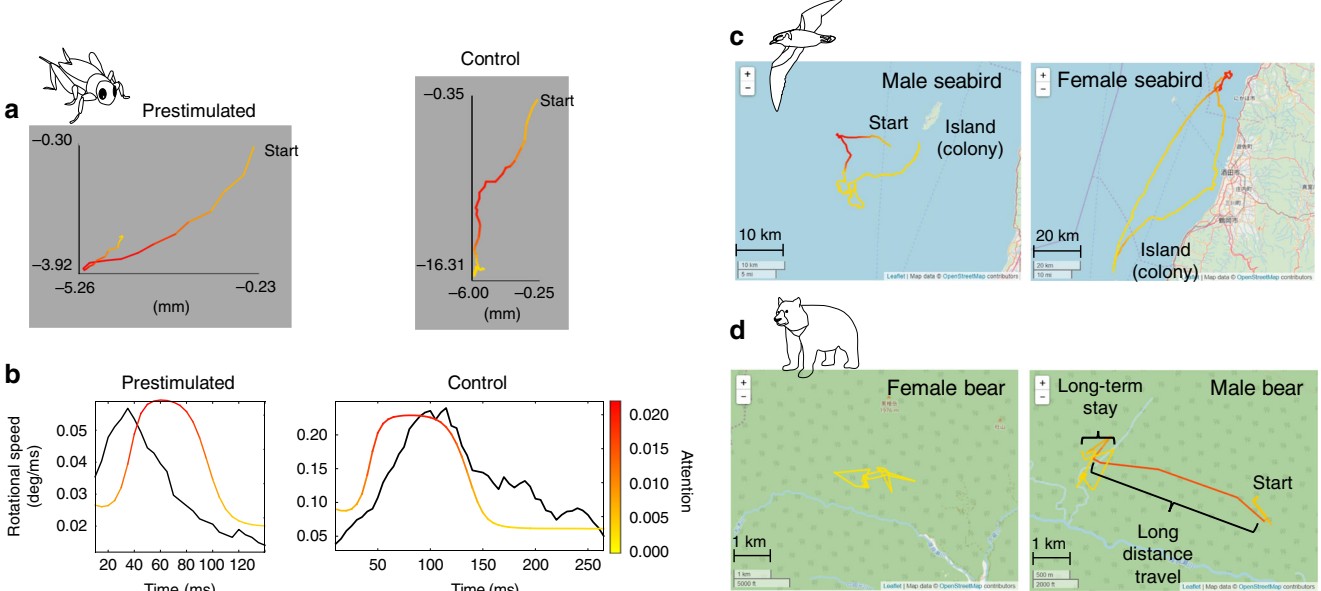

**Fig. 6 Highlighted trajectories of crickets, seabirds, and bears. a** Trajectories of the prestimulated and control crickets highlighted by DeepHL (see Supplementary Information, Application to the study of crickets, for more detail). **b** Time series of rotational speed (black lines) and attention value (colored lines) of the trajectories in **a**. A discriminator layer pays attention to the difference in the peaks of the rotational speed for the prestimulated and control classes. The high rotational speed was sustained in the control trajectory, which means that the crickets exhibited longer and larger turning movements in the control group. In contrast, the rotational speed transiently elevated and peaked earlier in the prestimulated group. **c** Trajectories of female and male seabirds highlighted by DeepHL. A discriminator layer pays attention to female trajectories when the locations of the female birds are close to the coastline (see Supplementary Information, Application to the study of seabirds). **d** Trajectories of female and male bears highlighted by DeepHL. The red segment is characteristic of the male bears, as detected by DeepHL. A discriminator layer pays attention to male trajectories when a male bear has traveled a long distance after/before staying in one place (see Supplementary Information, Application to the study of bears). Base map and data copyright OpenStreetMap contributors (License: www.openstreetmap.org/copyright).

synthetic trajectory data, see Supplementary Information, Evaluation with synthetic data). This versatile trajectory analysis was possible because of the useful functions of DeepHL. Furthermore, we confirmed that DeepHL does not require a large number of trajectories to train DeepHL-Net (Supplementary Table 5).

Discovering high-level features hidden in temporal dynamics, for example, the frequencies of worm movement speeds and the sustained rotation speed of crickets found with the help of DeepHL, is difficult in classic machine learning without an algorithm specifically designed for each task using prior knowledge gained by manual analysis, which requires substantial effort. In fact, this finding related to the worms has not been discovered by prior study[17] mainly performed by specialists who have been intensively studying the behavior of worms based on a classic approach, which comprehensively extracts 333 handcrafted locomotion features, on the worm data that are also used in our

study even though the finding of our study was obtained from the discriminator layer with the highest score. In addition, DeepHL was able to help in finding a prominent mouse movement feature related to exploration, which has not been a focus of prior studies and also obtained from a discriminator layer with the highest score. The discovered movement feature outperformed high-level features found in prior studies in terms of feature importance. This result is surprising because movement features of PD mice have been intensively studied by neuroscientists[23,24]. While the movement features of some animals such as seabirds and mice found with the help of DeepHL seem to be simple, the fact that these simple features have not been discovered after many years of research indicates the value of the findings given the difficulties of big behavioral data analysis based on classic approaches. Refer to Supplementary Information, Comparison with classic approaches, for analysis of the six animal species using classic

approaches. Whereas the classification accuracy of a classic approach is not extremely different from that of DeepHL, as shown in Supplementary Information, Comparison with classic approaches, it was difficult to find fine-grained characteristics of animal behaviors by using the classic approach because it employs only high-level features prepared in advance extracted from a whole trajectory.

In this study, we find a discriminator layer that focuses on a part of trajectory. However, it is possible for many attention layers to focus on the full trajectory. In such a case, we can assume that the global features are important for classifying the trajectories. We believe that such global features can be easily identified through classic statistical techniques or manual analysis.

Trajectory data observed from wild animals can include different noises. For example, the trajectory data from bears are noisy because of the forest canopy, as shown in Fig. 6d. When noises are included in the GPS measurements of both the male and female bears uniformly, we believe that DeepHL-Net can extract useful high-level features from the data. However, such noises can also degrade the classification performance. One possible solution to addressing this problem is to introduce a denoising autoencoder[33] (e.g., reducing noises during the preprocessing). In addition, the seabird GPS data include few sudden large errors. We can remove such errors by thresholding calculated speed (see Supplementary Information, Application to the study of seabirds). Moreover, GPS signals can be lost for a moment. However, primitive features used in this study, that is, speed and relative angular speed, are robust against such missing measurements.

There are several deep learning studies related to our work. Endo et al.[34] visualize/generate typical trajectories for taxis using an autoencoder. In addition, several visualization tools for interpreting the behavior of LSTMs have been developed, although these mainly focus on natural language processing[35,36]. LSTMs have also been used to predict worm trajectories[37], although these studies do not focus on comparative analysis. Recent deep learning studies have also employed attention mechanisms to visualize distinguishing features[38,39]. Attention mechanisms have also been actively studied in the computer vision field. Xu et al.[40] generated captions for an image by leveraging the attention of the input image to identify an important region in the image and generate each word. Zhang et al.[41] leveraged attention mechanisms to focus on foreground regions to alleviate distractions from the background for image-based salient object detection tasks. Park et al.[42] employed an attention mechanism to identify important regions in an image as well as generate textual descriptions using an LSTM for an image classification task. In the biology domain, Heras et al.[43] leveraged a deep attention network that predicts future turns of a zebrafish in a collective to identify surrounding zebrafish that affect the future turning of the focal zebrafish. Unlike in the above studies, in the present study, deep attention networks have been used to find distinguishing group-specific patterns in the trajectories.

Because the manual analysis of behavioral data is impractical for big behavioral data, we suggest that we are now on the cusp of changing the methods used for big data in biology research. We envision that DeepHL will transform the hitherto standard approach to comparative analysis from a hypothesis-driven approach, which relies on individual experience or manual analysis by researchers, to a data-driven approach. Owing to the useful functions proposed in this study, DeepHL enables researchers to easily extract insightful information hidden within a DNN that is trained on big data. Furthermore, because DeepHL is simply designed to find distinguishing trajectory segments between two groups, it can also be applied to a variety of

comparative analyses, for example, old vs. young animals, free-ranging vs. captive animals, animals from different habitats, animals with different life-history stages, food storing vs. non-storing individuals/species, social vs. solitary individuals/species, and specialists vs. generalists. Although many functions of DeepHL are tailored to a trajectory analysis, DeepHL-Net can process any type of time-series data. As a part of a future study, we plan to apply our network to other time series such as sounds emitted by animals.

We believe that these animal behavior analyses contribute not only to biology research, but also to the sustainable development of our society and coexistence with wild animals by understanding animal behavior. In addition, livestock farming has many potential applications of our animal behavior analysis. For example, our method can be applied to identifying characteristic behaviors of disease animals, productive cows, and submissive cows in a social hierarchy. Moreover, because trajectory data are observed from any moving objects, DeepHL is capable of wide application. Specifically, we believe that DeepHL can also be applied to trajectory analyses for humans and automobiles, which can contribute to our society in various aspects: improvement of work efficiency (e.g., analyzing trajectories of workers in logistics centers), healthcare (e.g., comparing between patients and healthy subjects), and eco-safe driving.

## Methods

**DeepHL system architecture**. The DeepHL system consists of three server computers. The first one is a web server that receives a trajectory data file from a user and provides analysis results to the user (Intel Xeon E5-2620 v4, 16 cores, 32 GB RAM, Ubuntu 14.04). The second one is a storage server that stores data files and analysis results. The third one is a GPU server that analyzes data provided by the user (Intel Xeon E5-2620 v4, 32 cores, 512 GB RAM, four NVIDIA Quadro P6000, Ubuntu 14.04). Supplementary Information, Algorithm, provides a complete description of the DeepHL method. DeepHL is accessible on the Internet through http://www-mmde.ist.osaka-u.ac.jp/maekawa/deephl/. Supplementary Information, User guide to DeepHL, provides a user guide to DeepHL. In addition, Supplementary Information, Usage of Python-based Software, and Supplementary Software 1 present the Python code of DeepHL.

**Preprocessing**. An input trajectory is a series of timestamps and $X/Y$ coordinates associated with a class label. To perform position- and rotation-independent analysis, we convert the series into time series of speed and relative angular speed and then standardize them (Supplementary Information, Algorithm). Note that the absolute coordinates of wild animals, which can relate to the distance from a nest or feeding location, for example, are important in understanding behavior of the animals. Hence, DeepHL allows the original coordinates to be input to DeepHL-Net along with the speed and relative angular speed. In addition, other biological time-series sensor data measured by the user can be fed into DeepHL-Net when these time-series data are included in a data file uploaded by the user. For example, a time series of the heading direction of animals obtained from digital compasses can be useful for behavior understanding. Moreover, primitive features usually used in trajectory analysis can be easily fed into DeepHL-Net. DeepHL automatically computes the travel distance from the initial position, the straight-line distance from the initial position, and the angle from the initial position (Supplementary Table 1) as primitive features. Using the web interface of DeepHL, the user can easily select primitive features and other sensor data to be fed into DeepHL-Net (Supplementary Information, User guide to DeepHL). See Supplementary Information, Effect of input features, for effects of input features on classification accuracy. Normally, the inputs of DeepHL-Net are two-dimensional time series, that is, speed and relative angular speed. When we input an additional time series (such as the original coordinates) into DeepHL-Net, the additional time series are added as additional dimensions of the inputs.

**Multi-scale layer-wise attention model (DeepHL-Net)**. Here, we explain DeepHL-Net shown in Fig. 2f in detail. The input of the model is a time series of primitive features, that is, an $l_{MAX} \times N_f$ matrix, where $l_{MAX}$ is the maximum length of the input trajectories and $N_f$ is the dimensionality of the time series, that is, the number of the primitive features. Because the lengths of observed trajectories are not identical to each other in many cases, we fill in missing elements in the matrix with $-1.0$ and mask them when we train DeepHL-Net. In each 1D convolutional layer of the convolutional stacks, we extract features by convolving input features through the time dimension using a filter with a width (kernel size) of $F_t$. We use different filter widths in the four convolutional stacks (3%, 6%, 9%, and 12% of

$l_{MAX}$) to extract features at different levels of scale. We use a stride (step size) of one sample in terms of the time axis. We also use padding to allow the outputs of a layer to have the same length as the layer inputs. In addition, to reduce an over-fitting, we employ a dropout, which is a simple regularization technique in which randomly selected neurons are dropped during training[44]. The dropout rate used in this study is 0.5.

In each LSTM layer of the LSTM stacks, we extract features considering the long-term dependencies of the input features. LSTM is a recurrent neural network architecture with memory cells, and it permits us to learn temporal relationships over a long time scale. LSTM learns long-term dependencies by employing memory cells that hold past information, updating the cell state using write, read, and reset operations with input, output, and forget gates (see Supplementary Information, Algorithm). In addition, we employ dropout to reduce overfitting. The attention information of each layer is computed by using Eq. (1), and then it is multiplied by the layer output. Here, the softmax and tanh functions in Eq. (1) are defined as follows:

$$\mathrm{softmax}\,(x_j) = \frac{\exp(x_j)}{\sum_i \exp(x_i)}, \qquad (2)$$

$$\tanh(x_j) = \frac{\exp(x_j) - \exp(-x_j)}{\exp(x_j) + \exp(-x_j)}. \qquad (3)$$

Note that parameters in Eq. (1) for each layer, that is, $W_a$ and $b_a$, as well as parameters in the convolutional and LSTM layers are estimated during the network training phase. Here, we introduced the tanh activation function into Eq. (1) to smooth out the output attention values. When an outlying large value is included in $W_a Z^T + b_a$ at time $t$, attention values other than time $t$ become extremely small without using the tanh function. When we visualize a trajectory using such attention values, only a single data point is colored in red, making it difficult for a user to identify important segments.

**Training and testing of DeepHL-Net.** The DeepHL user can select the parameters of DeepHL-Net used in the analysis, that is, the number of convolutional/LSTM layers and the number of neurons in each layer (default: four layers with 16 neurons). Then, DeepHL-Net is trained on 80% of randomly selected trajectories to minimize the binary classification error of the training data, employing back-propagation based on Adam[45] (Supplementary Information, Algorithm). (Note that each trajectory has a class label for binary classification.) Then, the trained DeepHL-Net is tested using the remaining 20% of trajectories to compute the classification accuracy, providing an indication of the degree of difference between the two classes.

**Computing the score of each layer.** To screen the layers in DeepHL-Net, we compute a score for each layer according to Eq. (4)

$$s(A_{i,C_A}, A_{i,C_B}) = s_{fc}(A_{i,C_A}, A_{i,C_B}) + s_{it}(A_{i,C_A}, A_{i,C_B}). \qquad (4)$$

Here, $A_{i,C_A}$ is a set of attention vectors calculated from trajectories belonging to class A using the $i$th layer. In addition, $A_{i,C_B}$ is a set of attention vectors calculated from trajectories belonging to class B using the $i$th layer. As mentioned in the main text, an attention vector from a discriminator layer should have large values within limited segments. Therefore, $s_{fc}(A_{i,C_A}, A_{i,C_B})$ in Eq. (4) calculates the averaged variance of the attention values normalized by the average length of the trajectories, as described in Eq. (5). When the layer focuses on a part of a trajectory, the variance increases

$$s_{fc}(A_{i,C_A}, A_{i,C_B}) = \sqrt{\frac{1}{|A_{i,C_A} \cup A_{i,C_B}| \cdot l(A_{i,C_A} \cup A_{i,C_B})} \sum_{\mathbf{a} \in A_{i,C_A} \cup A_{i,C_B}} V(\mathbf{a})}. \qquad (5)$$

Note that $V(\cdot)$ calculates the variance and $l(\cdot)$ calculates the average length of the trajectories. We take the square root of the average variance to derive the average standard deviation. Using $l(A_{i,C_A} \cup A_{i,C_B})$, which calculates the average length of $A_{i,C_A} \cup A_{i,C_B}$, we normalize the computed variance. Because the softmax function in Eq. (1) ensures that all values sum to 1, resulting in a larger variance for longer trajectories, we normalize the average variance using the average length.

In addition, as mentioned in the main text, the distribution of attention values by the layer for one class should be different from that for another class. Therefore, $s_{it}(A_{i,C_A}, A_{i,C_B})$ calculates the difference between the distributions of the attention values of classes A and B as follows:

$$s_{it}(A_{i,C_A}, A_{i,C_B}) = (1 - \mathrm{Intersect}\,(h(A_{i,C_A}), h(A_{i,C_B}))). \qquad (6)$$

Here, $h(\cdot)$ calculates a normalized histogram of attention with 200 bins, and Intersect($\cdot, \cdot$) calculates the area overlap between two histograms, and is described as follows:

$$\mathrm{Intersect}\,(H_1, H_2) = \sum_i \min(H_1(i), H_2(i)), \qquad (7)$$

where $H_1(i)$ shows the normalized frequency of the $i$th bin of histogram $H_1$.

As described in Eq. (4), the final score is calculated as the sum of the two scores of $s_{fc}(A_{i,C_A}, A_{i,C_B})$ and $s_{it}(A_{i,C_A}, A_{i,C_B})$.

Here, $s_{fc}(A_{i,C_A}, A_{i,C_B})$ in Eq. (4) is used to find a layer that focuses only on a portion of a trajectory. Owing to the term, only a small important portion of trajectories is highlighted in many cases, as shown in Figs. 3, 5, and 6, especially for the trajectories of beetles. However, substantial portions of several trajectories of the normal mice are highlighted, as shown in Fig. 4d. Because the characteristics of the normal mouse trajectories are the distance from the initial position, the segments in the trajectories far from the initial position are highlighted.

**Computing the correlation between attention values and handcrafted features.** To help the user understand the meaning of the highlights, DeepHL automatically computes the Pearson correlation coefficients between the attention values of each layer and handcrafted features computed by DeepHL, as shown in Supplementary Table 1. In addition, the correlation coefficients with sensor data and handcrafted features included in a trajectory data file are automatically computed. Computing the correlation with environmental sensor data can reveal the relationship between a behavior and environmental conditions. If a specific behavior is exhibited only when the temperature is high, for example, we can infer that the behavior relates to the high temperature condition. Furthermore, DeepHL automatically computes the moving average, moving variance, and derivative of each of the above features/sensor data, and then computes the correlation coefficients with the attention values, which are presented to the user (Supplementary Fig. 1).

**Computing the difference between distributions of each handcrafted feature for the two classes within highlighted segments.** To help the user understand the meaning of the highlights, DeepHL automatically computes the difference between distributions of each handcrafted feature for two classes within highlighted segments. The difference is computed as follows:

$$\mathrm{diff}(A_{i,C_A}, F_{j,C_A}, A_{i,C_B}, F_{j,C_B}) = 1 - \mathrm{Intersect}\,(h(m(A_{i,C_A}, F_{j,C_A})), h(m(A_{i,C_B}, F_{j,C_B}))). \qquad (8)$$

Here, $F_{j,C_A}$ is a set of time series of the $j$th handcrafted feature calculated from trajectories belonging to class A. In addition, $m(\cdot, \cdot)$ is a masking function that extracts feature values within highlighted segments. Because the softmax function in each attention layer ensures that all attention values in a sum of 1, we consider an attention value larger than $c/(\#\,\mathrm{time\,slices})$ as a potential attended value ($c = 1.2$ in our implementation).

**Data acquisition of worms.** Data acquisition was performed according to Yamazoe-Umemoto et al.[22]. In brief, several worms were placed in the center of an agar plate in a 9-cm Petri dish, 30% 2-nonanone (v/v, EtOH) was spotted on the left side of the plate, which was covered by a lid and placed on the bench upside down. Then, the images of the plate were captured with a high-resolution USB camera for 12 min at 1 Hz. Because the worms do not exhibit odor avoidance behavior during the first 2 min because of the rapid increase in odor concentration[46], the data for the following 10 min (i.e., 600 s) was used. From the images, individual worms were identified and the position of the centroid was recorded by an image processing software Move-tr/2D (v. 8.31; Library Inc., Japan). The number of recorded trajectories is 325 (Supplementary Table 2). The comparison was between the naive worms (control class) and the worms after preexposure to the odor (preexposed class).

**DeepHL analysis of worms.** A multivariate time series of movement speed, relative angular speed, distances from the initial position, and angle from the initial position extracted from the time series of trajectories was fed into DeepHL-Net, yielding a binary classification accuracy of 93.9%, where 20% of the data are used as test data. The discriminator layer used in this investigation has the highest score of all layers. As shown in Fig. 3d, which was calculated from the moving variance of the speed within highlighted segments, we can state that the changes in the speed of preexposed worms is larger than those of control worms. Figure 3e shows spectrograms of the speed calculated from entire trajectories (Fig. 3c) with a 128-s wide sliding window shifted in 1-sample intervals. In addition, Fig. 3f shows histograms of the dominant frequency of speed calculated from entire trajectories using the 128-s wide sliding window shifted in 1-sample intervals. These results also indicate the difference in the frequency of speed between the preexposed and control worms. Our investigation revealed that the dominant frequency of speed significantly differs between the preexposed and control worms using GLMM with Gaussian distributions ($t = -6.60$; d.f. $= 322.8$; $p = 1.68 \times 10^{-10}$, effect size($r^2$) $= 0.232$). The $p$ value is two sided. Individual factors were treated as random effects. The number of data points for the control class is $n = 76,784$ and that for the preexposed class is $n = 75,750$. We used GLMM with Gaussian distributions because the objective variable has a continuous value and we used the lmerTest package (v. 2.0–36) of R (v. 3.4.3) for the analysis.

**Data acquisition of mice.** We collected 52 trajectories of normal mice and unilateral 6-hydroxydopamine (OHDA) lesion mouse models of PD while they

freely moved for 10 min in an open field ($60 \times 55$ cm$^2$, wall height = 20 cm; normal: 22, PD: 30). The trajectories were detected by the animal's head position, which was captured by an overhead digital video camera (60 fps). Two sets of small red and green light-emitting diodes were mounted above the animal's head so that it could be located in each frame. Custom softwares based on Matlab (R2018b, Mathworks, MA, USA) and LabVIEW (Labview 2018, National Instruments, TX, USA) were used for tracking. We then created 30-s segments by splitting each trajectory because training a DNN requires a number of trajectories. We used 966 segments in total (normal: 374, PD: 592) collected from nine C57BL/6J mice (normal: 5, PD: 4). Note that we excluded 30-s segments that contain no movements of a mouse.

**DeepHL analysis of mice.** Movement speed, relative angular speed, travel distances, straight-line and travel distances from the initial position, and angle from the initial position were fed into our model. The accuracy for the binary classification of normal and 6-OHDA model mice was 74.7%, where 20% of the data are used as test data. The score of the discriminator layer was the highest of all LSTM layers and the sixth highest of all layers. Our investigation revealed that the behavior of visiting locations far away from the initial position can be characteristic of normal mice.

To evaluate PD symptoms from animal behaviors, previous studies have exclusively focused on the movement speed of animals in the open-field tests (frequency and bout duration of ambulation as well as immobility or fine movement) because typical symptoms in the animal model of PD are thought to be slowness of movement and a paucity of spontaneous movements. As shown in Fig. 4e–g, we found significant differences in average movement speed during ambulation periods, average movement speed during fine movement periods, and average maximum distance within a ±60-s window in a session. These differences are derived from the findings of DeepHL using the two-sided Wilcoxon rank-sum test ($W = 544$, $p = 3.486 \times 10^{-5}$, effect size (Cliff's delta) = $-0.648$; $W = 511$, $p = 5.869 \times 10^{-4}$, effect size (Cliff's delta) = $-0.548$; $W = 521$, $p = 2.666 \times 10^{-4}$, effect size (Cliff's delta) = $-0.579$). The 95% confidence intervals are [1.222, 3.481], [0.139, 0.468], and [13.726, 43.175], respectively. We used the exactRankTests package (v. 0.8–29) of R (v. 3.2.3). Note that these behavioral features are extracted from original 10-min trajectories.

The maximum distance, which was derived from a finding of DeepHL, is more useful for evaluating the PD symptoms than conventional measures based on the movement speed. Note that the new feature is designed based on an insight drawn from an analysis by deep learning. These results suggest that DeepHL helps find a novel measure not directly linked to the movement speed, that is, a straight-line distance within a certain time window. When the aim of an animal is to visit all locations in an area, the travel distance over a short duration commonly becomes longer. Besides, it is well known that rodents, including mice and rats, spontaneously prefer to explore an environment, particularly in novel places. Thus, DeepHL may capture the fact that the abnormal behavior of the 6-OHDA lesion model of PD hinders such spontaneous behavioral traits of normal mice. Indeed, the 6-OHDA lesion mouse model appears to remain in the same place. Although this hypothesis should be verified based on the causality between behavioral traits and neural activity patterns underlying PD symptoms using neuronal recording together with its optogenetic manipulation in the basal ganglia and motor cortex[23], it is beyond the scope of this study.

**Behavioral features of mice.** According to Kravitz et al.[23], ambulation was defined as periods when the velocity of the animal's center point averaged >2 cm/s for at least 0.5 s. Immobility was defined as continuous periods of time during which the average change of the trajectory was <1 cm for at least 1 s. Fine movement was defined as any movement that was not ambulation or immobility. Maximum travel distance within a ±60-s window was defined as the maximum straight-line distance between the center of the window and each point within the window. Note that each feature value is computed for each entire 10-min trajectory.

**6-OHDA injection of mice.** Under isoflurane anesthesia, 6-OHDA (4 mg/ml; Sigma) was injected through the implanted cannulae (AP −1.2 mm, ML 1.1 mm, DV 5.0 mm, 2 μl). Animals were allowed to recover for at least 1 week before post-lesion behavioral testing.

**Histological verification of dopaminergic cell loss.** After the mice were sacrificed by pentobarbital sodium overdose and perfused with formalin, their brains were frozen and cut coronally at 30 μl with a sliding microtome. For immunostaining, sections were divided into six interleaved sets. Immunohistochemistry was performed on the free-floating sections. Sections were pretreated with 3% hydrogen peroxide and incubated overnight with primary antibody mouse anti-tyrosine hydroxylase (1:1000; Millipore). As a secondary antibody, we used biotinylated donkey anti-mouse IgG (1:100; Jackson ImmunoResearch Inc.), followed by incubation with avidin–biotin–peroxidase complex solution (1:100; VECTASTAIN Elite ABC STANDARD KIT, Vector Laboratories). The immunoreactivities were

visualized by 3-3′ diaminobenzidine tetrahydrochloride (Dojindo Laboratories). The degree of dopaminergic cell loss was estimated by dividing the number of cells manually counted across three sections of the SNc (most rostral, most caudal, and the intermediate between them) of the lesioned hemisphere from that of the non-lesioned hemisphere.

**Data acquisition of beetles.** In the present study, we analyzed 419 walking trails collected from S- and L-strain beetles freely moving on a treadmill[47] (tracking software: custom software based on OpenCV, v. 2.4.9) using DeepHL (Supplementary Table 2). The number of the S-strain (L-strain) beetles is 20, consisting of 10 males and 10 females. The sampling rate of the treadmill is ~14.3 Hz, and the average duration of the trajectories is 52 s.

**DeepHL analysis of beetles.** In addition to the movement speed and relative angular speed, the distances and angle from the initial position were fed into DeepHL-Net. The classification accuracy for the binary classification between S and L strains was 84.5%, where 20% of the data are used as test data. The score of the discriminator layer in Fig. 5b was the third highest of all layers. Because the layer seems to focus on turns, we computed an angle of a trajectory segment for each point according to Fig. 5e. Figure 5f shows the average angles for the S and L strains (the number of data points for the S strain is 185,884 and the number of data points for the L strain is 219,497). We found that the angle for the S strain is significantly larger than that for the L strain, indicating that beetles of the S-strain beetles walk with more angle changing. Note that we used two-sided analysis of variance (ANOVA) ($F = 12.57$; d.f. = 1; $p = 0.001$; effect size($\eta^2$) = 0.09). Because multiple data points were computed from each individual beetle's trajectory, we treated the individuals as a random factor. The 95% confidence interval is [0.05, 0.18]. We used JMP 12.2.0., SAS. This result could indicate the difference in strategies for survival between the S- and L-strain beetles. The L-strain beetles can survive because of their long duration of TI against predators. In contrast, the S-strain beetles attempt to escape from a predator by frequently changing their moving directions.

Previous studies have shown a significantly lower expression level of brain dopamine in the beetles derived from the L strain than those from the S strain[30]. Nishi et al.[48] showed that injection of caffeine, which activates dopamine, decreases the duration of immobility in the L strain of *T. castaneum*. These phenomena concerning dopamine show an analogy to PD, which alters walking patterns[49]. In many animals, dopamine expression level relates to movement patterns, and a specific trajectory segment pattern of the L strain might be similarly deeply affected. To test this new hypothesis, the relationship between dopamine expression and detailed analysis for walking ability, which should be done apart from the present study, should be examined in the future. In conclusion, the analysis using DeepHL revealed significantly different walking trajectories between beetles from the S and L strains using ANOVA: the S-strain beetles walk with more angle changes along the direction of travel compared to the L-strain beetles.

**Ethics statement.** The studies on streaked shearwaters, mice, and bears were approved by the Animal Experimental Committees of Nagoya University (streaked shearwaters), the Doshisha University Institutional Animal Care and Use Committees (mice), and the Institutional Animal Care and Use Committee of Tokyo University of Agriculture and Technology (bears), respectively. The research on streaked shearwaters was conducted with permits from the Ministry of the Environment, Japan. All experimental procedures used in the bear research followed the Guidelines Concerning Animal Experimentation of the Tokyo University of Agriculture and Technology and the Mammal Society of Japan. They specify no requirements for the treatment of insects in experiments. Details of animals used in this study are described in Supplementary Information, Animals.

**Reporting summary.** Further information on research design is available in the Nature Research Reporting Summary linked to this article.

## Data availability

The dataset of the worms analyzed during the current study is available in the Dryad repository, https://doi.org/10.5061/dryad.37pvmcvf5, and included in Supplementary Data 1. The datasets of the mice, beetles, crickets, and seabirds analyzed during the current study are included in Supplementary Data 1. The dataset of the bears are available from the corresponding author upon reasonable request because the release of the bear data can increase the likelihood of poaching and stir up the fear in residents. Source data are provided with this paper.

## Code availability

The source code of DeepHL is distributed as Supplementary Software 1. The most recent version of the software is available at https://doi.org/10.5281/zenodo.402393150. The use of the software is exclusively limited to the purpose of undertaking academic, governmental, or not-for-profit research. The DeepHL web system is accessible on the

Internet through http://www-mmde.ist.osaka-u.ac.jp/maekawa/deephl/. We will keep the website operating and freely accessible for the foreseeable future.

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

## Acknowledgements
We thank Rory P. Wilson for comments on the manuscript. We are also grateful to Chinatsu Kozakai, Tomoya Abe, Masahiro Ogawa, and Maki Yamamoto for their field assistance. This work was supported by JSPS Kakenhi JP16H06539, JP16H06545, JP16H06544, JP16H06543, JP16H06541, JP17H05976, and JP17H05971.

## Author contributions
T.Maekawa conceived and directed the study, and performed method design, software implementation, data analysis, and manuscript writing. K.O. and Y.Z. performed method design, software implementation, and data analysis. K.D.K., S.J.Y., K.Yoda, S.T., H.O., H.S., M.F., T.Miyatake, K.M., and S.K. performed data collection, data analysis, and manuscript writing. S.M., R.F., N.N., K.Yamazaki, and K.I. performed data collection and developed data collection devices.

## Competing interests
The authors declare no competing interests.

**Additional information**

