## [Peer Review File · Nature Communications]

REVIEWER COMMENTS

Reviewer #1 (Remarks to the Author):

The authors use deep learning with an attention mechanism to select segments of the trajectories that distinguish two groups A and B. The method proposed is, to our knowledge, novel and of interest to biologists. There is, however, literature about using attention layers to build explainable AI classification methods (e.g. Huk Park et al, 2018 and references) that maybe should be better acknowledged.

Methods and Supplementary were nice to read, but we found Results and the second half of the Introduction more difficult to read. I think a revision of this text would be very positive.

introduction:

The final paragraphs (after line 50) are way too detailed and disorganised. We propose to sum up the paper contribution in a single paragraph and leave most details for the following sections.

Results:

A paragraph summing up the method pipeline (with few details) at the beginning of results.

The explanation on how the layer-wise attention (around equation 1) is constructed is confusing. Mentioning the dimensions of each element explicitly, or saying that all convolutions are 1D convolutions, would help. For example, as we understand, in equation 1, b_a is a vector of 1 dimension, Z has 2 and W_a 1 and it would be best to say this explicitly.

It is not clear to us how the final classification output is produced from the attention outputs. How exactly this is done is critical to assess what it means that a trajectory segment has high or low attention and thus the validity of the method.

What is the rationale behind not studying the cases where attention spreads along a trajectory? Surely, if such attention outputs appear after training, the implication is that to classify trajectories both global and local features are important.

For each application, it would be good to see the classification accuracy on the test dataset.

Discussion:

Details:

50 and ss: We propose to give a name to the network used (DeepHLNet?) and stick to it. Currently, Deep Neural Network (DNN) is used. This is too general and causes problems. For example, in the next few lines, the authors make many statements about DNN, leaving the reader thinking which ones are about the network proposed and which ones are general statements about DNNs!

120: abstracted > abstract

121: "perspectives", and in other places "views". Maybe scales is a better word?

Eq 1 and line 129: Why a tanh before a softmax? Softmax outputs values between 0 and 1 even if the input is not restricted.

132-133: In figure 2 it is not indicated that the outputs are concatenated. In addition, it is not clear how they are "used to output an estimate". A feed-forward network?

146: Maybe the word "learned" is not a good choice. Perhaps, "preexposed" is better.

156 and ss.: In this paragraph "attention" is used both with its common meaning and with the technical meaning, potentially confusing the reader.

164 and ss.: Equation is confusing and some notation is explained too late. Maybe worth moving to Methods? To make it easier for the reader we would suggest explaining each term separately as

two scores, s_1 and s_2 . First, motivate s_1 , then formula of s_1 and explaining s_1 . Second, motivate s_2 , formula of s_2 and explaining s_2 . Finally a statement saying that the score is the sum of s_1 and s_2 .

174-177: This part needs rewriting

208: "is the highest". Among what?

260: That normal mice explore "unvisited places" is not implied but only suggested by the results.

314 Confusing opening sentence

417-420 Is filter width the same concept as kernel size? Do you use any padding?

426: There is a mention of dropout for LSTM, but no mention before in the convolutional layers. Do you use dropout there?

Equation 4: There is something missing (a minus sign?) after the 1

479 and ss. A discussion about the importance of research on *C. elegans* is beyond the scope of methods. Maybe results or discussion?

Missing references:

This is not the first architecture with attention mechanisms at different layers (e.g. Zhang et al 2018,). Please put your proposed architecture in the context of these works.

There is a large literature advocating the use of an attention mechanism to 'open the black box' of neural networks, particularly in the field of image caption generation (e.g. Xu et al 2015), and in the study of animal trajectories (e.g. Heras et al, 2019).

The authors may want to consider references on the use of attention mechanism to understand/study solution to a problem:

Heras, Francisco JH, et al. "Deep attention networks reveal the rules of collective motion in zebrafish." *PLoS computational biology* 15.9 (2019): e1007354.

Huk Park, Dong, et al. "Multimodal explanations: Justifying decisions and pointing to the evidence." *Proceedings of the IEEE Conference on Computer Vision and Pattern Recognition*. 2018.

Xu, Kelvin, et al. "Show, attend and tell: Neural image caption generation with visual attention." *International conference on machine learning*. 2015.

Zhang, Xiaoning, et al. "Progressive attention guided recurrent network for salient object detection." *Proceedings of the IEEE Conference on Computer Vision and Pattern Recognition*. 2018.

Reviewer #2 (Remarks to the Author):

The article propose a framework, named deepHL, to investigate animal behavior based on trajectory data. It aims at (i) automatically highlight statistical differences between trajectories of animals coming from two different groups (e.g. male vs female), (ii) try to explain these differences based on a list of handcrafted features.

For me, the major contribution of their framework is the definition and computation of the « attention ». It is automatically learned, based on a deep learning model, specifically designed for this application, to classify/differentiate between the trajectory of the two groups. It's the first time I'm seeing this type of approach and I consider it can be a major contribution in analysis of big data, but I also have to admit I don't necessarily have a deep knowledge on the literature on this topic.

Once attention is computed, some parts of the trajectory can be highlighted, these are the part that makes the trajectory from the 2 groups different. It is based on a scoring function that allows

determining which layers of the DL model are the more discriminant. When multiple trajectories are available, the framework automatically select trajectories containing the characteristic segments (which are learned by the model to discriminate between animals of the 2 groups).

Then, the framework propose a way to explain these characteristic segments, based on statistics that are interpretable by human. These statistics are called handcrafted features and they are inputs of the model. In some cases, the considered handcrafted features might not explain the difference, but being able to visualize these characteristic segments certainly help the researcher to define new features (as they show in their examples). Be careful as in couple of places, the article is written such that ne can think deepHL automatically create these features. Examples are :

* Line 45. « Although trajectory analysis based on classic machine learning has been studied, it still relies on features handcrafted by experienced researchers based on finding ... », we are still here, but with a powerfull, user friendly tool allowing to automatically choose among these feature on a large dataset. But also facilitate the work of the researcher, when the characteristic segments are known, he can think of how best they can be explained.

* Line 339. « DeepHL was able to find new prominent », should be help finding.

* Line 554. « DeepHL found a novel measure... », help finding

* More generally, I think the different definition of the features doesn't help (see below).

Other comments I have

Many trajectory data come from GPS for wild animal. The precision of this type of data can be pretty bad. For example, single can be lost for a moment, or the GPS precision can become very low which noised the trajectory. Is the model robust to GPS lack of precision?

I found sometimes the definition of features can be hard to follow . It needs to be clarified. It'll help distinguish between the output of the model, and the user inputs.

It seems that the handcrafted features are features defined by human, such that speed acceleration etc. Actually they are defined in Supp Table 1.

Then we have the primitive features, defined line 105, as speed and relative angular speed. So I guess it could simply be called handcrafted features.

Finally, we have the high-level features. When I read line 40, I could think this is again the handcrafted features, because it's a user input. But when I read line 58, no high level features are not handcrafted, they are automatically extracted but the DNN, and difficult to interpret. Line 74, high-level features are design by biologists.

Features also has its own meaning in CNN...

DeepHL seems to be designed for spatial trajectory. Can we use it with any time-series data?

It seems that environmental data can be included along the position. But it's not clear how? Maybe an example on this type of application can be beneficial. If no data is present, why not simulate a dataset, with for example different behavior between A and B when the temperature is changing?

This work can be useful to many fields, and maybe the authors could give more examples, which will help the diffusion of their work. I personally think of livestock farming. For example social network for domesticated animals, i.e. to determine hierarchy of the individual within a flock. Maybe in animal health and welfare, can we distinguish between diseased and non-diseased animals?

Paragraph starting line 115. You could maybe mentioned the color of the layers in Figure 2 when you introduce them.

Line 129. I guess it's not exactly equation 1 because of the softmax function. Softmax is not in the neuron definition?

Line 418, I wonder how well the deep learning model is, compared to other classical classifiers (e.g. SVM, random forest etc)? Same Line 509.

Line 428, 0.5 dropout. Fixed manually? Based on some tests? Can it be tuned?

Congratulation to the authors for their work!

DeepHL: Deep Learning-assisted Comparative Analysis of Animal Trajectories

We would like to take this opportunity to thank all of the reviewers for their constructive comments regarding our original submission, which have helped us in improving the quality of the paper. We would also like to thank you for your help throughout this process.

-----**Response for Reviewer #1**-----

[Condition 1]

introduction: The final paragraphs (after line 50) are way too detailed and disorganised. We propose to sum up the paper contribution in a single paragraph and leave most details for the following sections.

[Answer]

Thank you for your comment. According to the comment, we simplified the structure of the introduction section (after line 51) as follows.

1. We described our approach (leveraging a deep neural network for a comparative analysis).
2. We described the problem (a black box problem).
3. We described our solution (DeepHL with attention-based neural networks).

We believe that the readability of the introduction has been improved owing to the comments given.

(Lines 51–70)

In this study, we present a new data-driven approach based on deep learning to support an analysis by biologists, as illustrated in the lower part of Fig. 1a. Specifically, this study focuses on a comparative analysis, and a deep learning-based method is proposed to help identify the differences between the trajectory data of two groups. With this approach, to extract the high-level features from the trajectory data for a classification of the two groups, we leverage the feature learning capacity of deep learning, i.e., learning of the high-level feature extraction processes performed within a deep neural network (DNN), which was originally conducted by experienced researchers. Although a DNN can extract high-level features objectively, unlike a classic approach, a DNN is regarded as a black box, making it difficult to interpret the meaning of the high-level features learned by the DNN, i.e., to observe the group differences detected by the network. To address this problem, we developed DeepHL, a free, user-friendly, web-based software, in which an interpretable neural network with multi-scale layer-wise attention [14] is used to elucidate the characteristic segments in the trajectories to which the proposed DNN model focuses on in order to distinguish between the trajectories of the two groups (Fig. 1b,c). Because this method informs researchers regarding "which parts of the trajectories they should focus on for further analysis," researchers can save time and effort related to an otherwise manual analysis of huge numbers of trajectories to derive the characteristic segments. In addition, DeepHL finds handcrafted features prepared in advance that are highly correlated with the identified segments to help the researchers consider how best the segments can be explained. Thus, this method facilitates data-driven

research for a comparative analysis by supporting knowledge discovery from the data.

[Condition 2]

A paragraph summing up the method pipeline (with few details) at the beginning of results.

[Answer]

According to the comment, we added a summary description at the beginning of the results section.

(Lines 87–97)

Here, we briefly introduce the pipeline of the proposed method. (i) DeepHL first trains our proposed network (hereafter called DeepHL-Net) on the trajectory data from two classes. (ii) The attention mechanism in DeepHL-Net then calculates the attention value of each data point in each trajectory for each layer in DeepHL-Net. (iii) Once the attention values are computed, some parts of the trajectories are highlighted by DeepHL using the attention output from a particular layer that is assumed to capture differences in the two classes. To help a user of DeepHL find such a layer (hereinafter, a "discriminator layer") in DeepHL-Net, DeepHL calculates the score for each layer based on attention outputs from the layer. (iv) DeepHL also supports the user in explaining the meaning of the highlighted segments based on a list of handcrafted features from the trajectories prepared in advance by calculating the correlation between the attention values and each of the handcrafted features (Fig. 1d).

[Condition 3]

The explanation on how the layer-wise attention (around equation 1) is constructed is confusing. Mentioning the dimensions of each element explicitly, or saying that all convolutions are 1D convolutions, would help. For example, as we understand, in equation 1, \mathbf{b}_a is a vector of 1 dimension, \mathbf{Z} has 2 and \mathbf{W}_a 1 and it would be best to say this explicitly.

[Answer]

Thank you for the comment. We added information regarding the dimensions of the elements in Equation (1). We also specified that 1D convolutions are used.

(Please refer to lines 132–137)

[Condition 4]

It is not clear to us how the final classification output is produced from the attention outputs. How

exactly this is done is critical to assess what it means that a trajectory segment has high or low attention and thus the validity of the method.

[Answer]

Thank you for the comment. The attention outputs are multiplied by the outputs of the 1D convolutional/LSTM layer, and the multiplied outputs of all the layers are then concatenated to calculate the final classification output in the softmax output layer. We have clearly specified this fact. We have also modified Fig. 2f and the descriptions according to Comment 5 by Reviewer 1. Please also refer to our response to Comment 5.

In addition, the classification outputs of the test data are used to calculate the classification accuracy of the test data, which can be used to estimate whether the neural network has been properly trained. When the accuracy is close to 0.5, we can regard the network to have not been properly trained, indicating that there are no differences between the two classes or that the training data have problems (e.g., too few data are used). We added proper descriptions to the revised version of the manuscript. In our study, the classification accuracies of the worm, mouse, and beetle studies are 0.939, 0.747, and 0.845, respectively, which were described only in the Methods section. We have added information to the main text according to Condition 6 by Reviewer 1. Please refer also to our response to Condition 6.

Moreover, the DeepHL web interface provides a classification result for each trajectory, which is described in Supplementary Information. We recommend the user to ignore misclassified trajectories when the user browses trajectories. We added a description about it to the main text.

(Lines 141–144)

The multiplied outputs of all layers are concatenated and then used to output an estimate, i.e., class A or B, in a densely connected output layer using the softmax function, i.e., the final layer in DeepHL-Net (green-colored block in Fig. 2f).

(Lines 160–164)

Here, the classification accuracy of the trained DeepHL-Net was 93.9% (see Methods), indicating that DeepHL-Net was properly trained. When the accuracy is low, e.g., 50%, we can regard such a state as having no differences between the two classes or the training data having certain problems (e.g., an excessively small amount of data).

(Lines 198–200)

In addition, the DeepHL web interface provides a classification result for each trajectory, permitting the user to ignore misclassified trajectories when the user browses trajectories.

[Condition 5]

What is the rationale behind not studying the cases where attention spreads along a trajectory? Surely, if such attention outputs appear after training, the implication is that to classify trajectories both global and local features are important.

[Answer]

Thank you for the insightful comment. When all attention values are high (identical) in a trajectory, for example, we believe that it does not provide any useful information to the user because we are unable to identify the specific differences between the two groups (in this case, the entire trajectory is shown in a single color). However, as pointed out by the reviewer, when many attention layers focus on the entire trajectory, for example, we can assume that the global features are important. We have added a discussion about this in the revised version of the paper.

(Lines 378-381)

In this study, we find a discriminator layer that focuses on a part of trajectory. However, it is possible for many attention layers to focus on the full trajectory. In such a case, we can assume that the global features are important for classifying the trajectories. We believe that such global features can be easily identified through classic statistical techniques or manual analysis.

[Condition 6]

For each application, it would be good to see the classification accuracy on the test dataset.

[Answer]

Thank you for the comment. The accuracies of the classification are shown in the Methods section. However, according to the comment, we added information in the main text. Please refer also to our response to Condition 4.

(Lines 251–252)

The classification accuracy for the mouse dataset is 74.7% (see Methods for further details).

(Line 286)

The classification accuracy for the beetle dataset is 84.5% (see Methods for further details).

[Comment 1]

50 and ss: We propose to give a name to the network used (DeepHLNet?)

[Answer]

Thank you for the suggestion. We use “DeepHL-Net” in this study. Because we defined “DeepHL-Net” at the beginning of the results section, we used the term “DeepHL-Net” thereafter.

[Comment 2]

120: abstracted > abstract

[Answer]

Thank you for the comment. We have modified the sentence accordingly.

[Comment 3]

121: “perspectives”, and in other places “views”. Maybe scales is a better word?

[Answer]

Thank you for the comment. We decided to use “scales” instead of “perspectives” and “views.”

[Comment 4]

Eq 1 and line 129: Why a tanh before a softmax? Softmax outputs values between 0 and 1 even if the input is not restricted.

[Answer]

Thank you for the valuable comment. We have introduced the tanh function to smooth out the output attention values. When an outlying large value is included in $W_a Z^T + b_a$ at time t , we confirmed that attention values other than time t become extremely small without using the tanh function. When we visualize a trajectory using such attention values, only a single data point is colored in red, making it difficult for a user to identify important segments. We added a description regarding this in the revised version of the paper.

(Lines 490-495)

Here, we introduced the tanh activation function into Equation (1) to smooth out the output attention values. When an outlying large value is included in $W_a Z^T + b_a$ at time t , attention values other than time t become extremely small without using the tanh function. When we visualize a trajectory using such attention values, only a single data point is colored in red, making it difficult for a user to identify important segments.

[Comment 5]

132-133: In figure 2 it is not indicated that the outputs are concatenated. In addition, it is not clear how they are “used to output an estimate”. A feed-forward network?

[Answer]

Thank you for pointing out our mistake. We included a “Concatenate” block in Figure 2f. The concatenated outputs are then fed into the densely connected output layer to estimate a class label (i.e., feed-forward network). We added the following description about this to the revised version of the manuscript.

(Lines 141–144)

The multiplied outputs of all layers are concatenated and then used to output an estimate, i.e., class A or B, in a densely connected output layer using the softmax function, i.e., the final layer in DeepHL-Net (green-colored block in Fig. 2f).

[Comment 6]

146: Maybe the word “learned” is not a good choice. Perhaps, “preexposed” is better.

[Answer]

Thank you for the comment. We modified it to “preexposed.”

[Comment 7]

156 and ss.: In this paragraph “attention” is used both with its common meaning and with the technical meaning, potentially confusing the reader.

[Answer]

Thank you for the comment. We have clearly separated the general and technical meaning by introducing the phrase “Technically speaking...”

(Lines 170–173)

Technically speaking, an attention vector from the discriminator layer should have large values within limited segments. When the attention values are identical throughout the entire trajectory, the user cannot determine which part of the trajectory is characteristic of the class of interest.

(Lines 176–180)

Technically speaking, a distribution of attention values using the layer for one class should be different from that for another class. For example, when the layer exhibits large attention values to segments in trajectories belonging to only one class, the user can easily understand that these segments are characteristic of that class.

[Comment 8]

164 and ss.: Equation is confusing and some notation is explained too late. Maybe worth moving to Methods? To make it easier for the reader we would suggest explaining each term separately as two scores, s_1 and s_2 . First, motivate s_1 , then formula of s_1 and explaining s_1 . Second, motivate s_2 , formula of s_2 and explaining s_2 . Finally a statement saying that the score is the sum of s_1 and s_2 .

[Answer]

Thank you very much for your suggestion. According to the comment, we moved the main part related to the equation to the Methods section. In addition, we described the equation according to the suggested procedures in the same section.

(Line 181)

See the Methods section for an equation to calculate the score.

(Please also see lines 506–526)

[Comment 9]

174-177: This part needs rewriting

[Answer]

Thank you for the comment. We simplified the sentences to improve the readability.

(Lines 181–183)

The DeepHL web interface provides a ranking of the layers based on the calculated scores, enabling the user to easily find high-scoring layers, which can provide an insightful highlight of the trajectory.

[Comment 10]

208: “is the highest”. Among what?

[Answer]

Thank you for the comment. It is the highest among all handcrafted features. We modified the sentence as follows.

(Lines 215–217)

In the worm example, the absolute correlation coefficient between the attention values of the layer and the moving average of the worm speed is the highest among all handcrafted features (Supplementary Table 3).

[Comment 11]

260: That normal mice explore “unvisited places” is not implied but only suggested by the results.

[Answer]

Thank you for the comment. We modified the sentence according to the comment as follows.

(Line 270)

The results suggest that normal mice prefer exploring unvisited locations.

[Comment 12]

314 Confusing opening sentence

[Answer]

Thank you for the comment. This is the opening sentence of the paragraph summarizing the animal analysis.

We modified the sentence as follows.

(Lines 338–339)

In the above analysis of the worms, mice, insects, seabirds, and bears, we could discover new findings that were not revealed through a manual analysis or classic machine learning.

[Comment 13]

417-420 Is filter width the same concept as kernel size? Do you use any padding?

[Answer]

Yes. The filter width is the same as the kernel size. We also used padding to make the outputs of the layer have the same length as the inputs of the layer. We modified/added the following sentences.

(Line 475 (473–475))

In each 1D convolutional layer of the convolutional stacks, we extract features by convolving input features through the time dimension using a filter with a width (kernel size) of F_t .

(Lines 477–478)

We also use padding to allow the outputs of a layer to have the same length as the layer inputs.

[Comment 14]

426: There is a mention of dropout for LSTM, but no mention before in the convolutional layers. Do you use dropout there?

[Answer]

We also used a dropout for the convolutional layer, as shown in Fig. 2f. However, we did not describe its use. We therefore added the following sentence.

(Lines 478–480)

In addition, to reduce an overfitting, we employ a dropout, which is a simple regularization technique in which randomly selected neurons are dropped during training [43]. The dropout rate used in this study is 0.5.

[Comment 15]

Equation 4: There is something missing (a minus sign?) after the 1

[Answer]

Yes. There is a minus sign after the 1. We can see the minus sign when viewing the file with Adobe Acrobat X Pro, PDF-XChange viewer, SumatraPDF, Google Chrome, and Microsoft Edge. Here is a screenshot from Adobe Acrobat X Pro.

$$\text{diff}(A_{i,C_A}, F_{j,C_A}, A_{i,C_B}, F_{j,C_B}) = 1 - \text{Intersect}(h(m(A_{i,C_A}, F_{j,C_A})), h(m(A_{i,C_B}, F_{j,C_B})))$$

We hope that the reviewers can see the sign in the revised version of the paper (Equation 8 in p. 37).

[Comment 16]

479 and ss. A discussion about the importance of research on *C. elegans* is beyond the scope of methods. Maybe results or discussion?

[Answer]

We agree with your comment. We moved “Significance of analysis of worms” and “Significance of analysis of red flour beetles” to the Supplementary Information section.

(Lines 239–241)

The biological significance of the worm and other animal analyses are described in the Supplementary

Information.

[Comment 17]

The authors may want to consider references on the use of attention mechanism to understand/study solution to a problem:

- Heras, Francisco JH, et al. "Deep attention networks reveal the rules of collective motion in zebrafish." *PLoS computational biology* 15.9 (2019): e1007354.
- Huk Park, Dong, et al. "Multimodal explanations: Justifying decisions and pointing to the evidence." *Proceedings of the IEEE Conference on Computer Vision and Pattern Recognition*. 2018.
- Xu, Kelvin, et al. "Show, attend and tell: Neural image caption generation with visual attention." *International conference on machine learning*. 2015.
- Zhang, Xiaoning, et al. "Progressive attention guided recurrent network for salient object detection." *Proceedings of the IEEE Conference on Computer Vision and Pattern Recognition*. 2018.

[Answer]

Thank you for suggesting the related studies. We added the following sentences regarding the related studies.

(Lines 397–407)

Attention mechanisms have also been actively studied in the computer vision field. Xu et al. generated captions for an image by leveraging the attention of the input image to identify an important region in the image and generate each word [39]. Zhang et al. leveraged attention mechanisms to focus on foreground regions to alleviate distractions from the background for image-based salient object detection tasks [40]. Park et al. employed an attention mechanism to identify important regions in an image as well as generate textual descriptions using an LSTM for an image classification task [41]. In the biology domain, Heras et al. leveraged a deep attention network that predicts future turns of a zebrafish in a collective to identify surrounding zebrafish that affect the future turning of the focal zebrafish [42]. Unlike in the above studies, in the present study, deep attention networks have been used to find distinguishing group-specific patterns in the trajectories.

-----**Response for Reviewer #2**-----

[Condition 1]

Be careful as in couple of places, the article is written such that ne can think deepHL automatically create these features. Examples are :

- Line 45. « Although trajectory analysis based on classic machine learning has been studied, it still

relies on features handcrafted by experienced researchers based on finding ... », we are still here, but with a powerful user friendly tool allowing to automatically choose among these feature on a large dataset. But also facilitate the work of the researcher, when the characteristic segments are known, he can think of how best they can be explained.

- Line 339. « DeepHL was able to find new prominent », should be help finding.

- Line 554. « DeepHL found a novel measure... », help finding

[Answer]

Thank you for the comment. As mentioned by the reviewer, our method will help the user select meaningful handcrafted features that are highly correlated with highlighted segments. We simplified the introduction (according to Condition 1 by Reviewer 1) and clearly specify that the selected features can be used to describe the identified segments according to the comment. In addition, we modified sentences that may have confused the readers.

(Lines 67–69)

In addition, DeepHL finds handcrafted features prepared in advance that are highly correlated with the identified segments to help the researchers consider how best the segments can be explained.

(Lines 356–359)

Discovering high-level features hidden in temporal dynamics, e.g., the frequencies of worm movement speeds and the sustained rotation speed of crickets found with the help of DeepHL, is difficult in classic machine learning without an algorithm specifically designed for each task using prior knowledge gained by manual analysis, which requires substantial effort.

(Lines 364–366)

In addition, DeepHL was able to help in finding a new prominent mouse movement feature related to exploration, which has not been a focus of prior studies and also obtained from a discriminator layer with the highest score.

(Lines 369–372)

While the movement features of some animals such as seabirds and mice found with the help of DeepHL seem to be simple, the fact that these simple features have not been discovered after many years of research indicates the value of the findings given the difficulties of big behavioral data analysis based on classic approaches.

(Lines 613–614)

These results suggest that DeepHL helps find a novel measure not directly linked to the movement speed, i.e., a straight-line distance within a certain time window.

[Comment 1]

Many trajectory data come from GPS for wild animal. The precision of this type of data can be pretty bad. For example, single can be lost for a moment, or the GPS precision can become very low which noised the trajectory. Is the model robust to GPS lack of precision?

[Answer]

Thank you for the comment. The trajectory data from the bears are noisy because of the forest canopy, as shown in Fig. 6b. When noises are included in GPS measurements of both male and female bears uniformly, we believe that our neural network is sufficiently robust to extract useful features for classification from the data because we could obtain sex-dependent behavioral features of the bears as shown in Fig. 6d. However, the noises can also degrade the classification performance by the network, as pointed out by the reviewer. One possible solution to address this problem is introducing a denoising autoencoder. In addition, the seabird GPS data include few sudden large errors. To remove the errors, we first computed the movement speed for each data point and then simply discarded data points with speeds faster than 80 km/h, which is the upper bound for birds, as described in Supplementary Information. Moreover, as pointed out by the reviewer, GPS signals can be lost for a moment. However, because we employ speed-related primitive features as neural network inputs, we believe that they are robust against missing measurements.

We added a description about GPS errors to the Discussion section.

(Lines 382–391)

Trajectory data observed from wild animals can include different noises. For example, the trajectory data from bears are noisy because of the forest canopy, as shown in Fig. 6b. When noises are included in the GPS measurements of both the male and female bears uniformly, we believe that DeepHL-Net can extract useful high-level features from the data. However, such noises can also degrade the classification performance. One possible solution to addressing this problem is to introduce a denoising autoencoder [32] (e.g., reducing noises during the preprocessing). In addition, the seabird GPS data include few sudden large errors. We can remove such errors by thresholding calculated speed (see Supplementary Information, *Application to the study of seabirds*). Moreover, GPS signals can be lost for a moment. However, primitive features used in this study, i.e., speed and relative angular speed, are robust against such missing measurements.

[Comment 2]

I found sometimes the definition of features can be hard to follow. It needs to be clarified. It'll help distinguish between the output of the model, and the user inputs.

It seems that the handcrafted features are features defined by human, such that speed acceleration etc. Actually they are defined in Supp Table 1.

Then we have the primitive features, defined line 105, as speed and relative angular speed. So I guess it could simply be called handcrafted features.

Finally, we have the high-level features. When I read line 40, I could think this is again the handcrafted features, because it's a user input. But when I read line 58, no high level features are not handcrafted, they are automatically extracted but the DNN, and difficult to interpret. Line 74, high-level features are design by biologists.

Features also has its own meaning in CNN...

[Answer]

We apologize for the complicated definitions. A high-level feature is a feature that represents the characteristics of the entire trajectory and can be designed/acquired by researchers or deep neural networks. Handcrafted features are low-level features handcrafted by researchers and include primitive features (speed and relative angular speed). We clearly specified that high-level features are also acquired by deep neural networks. In addition, we summarized the definitions of these features at the beginning of the Results section.

(Lines 54–58)

With this approach, to extract the high-level features from the trajectory data for a classification of the two groups, we leverage the feature learning capacity of deep learning, i.e., learning of the high-level feature extraction processes performed within a deep neural network (DNN), which was originally conducted by experienced researchers.

(Lines 98–106)

Before describing our method in detail, herein we provide definitions of the features used in this study. Primitive features are basic features widely used in a locomotion analysis, i.e., speed and relative angular speed. Handcrafted features are low-level features handcrafted by researchers, such as acceleration and distance from the initial position, and include the primitive features. High-level features are designed by researchers and characterize a group through a comparative analysis. A high-level feature is computed from the entire trajectory, such as the average movement speed and duration of stay at a feeding location. Although a DNN can also acquire high-level features or concepts, we found it difficult to comprehend these high-level features. A feature calculated in each layer in the DNN is simply called a feature.

[Comment 3]

DeepHL seems to be designed for spatial trajectory. Can we use it with any time-series data?

[Answer]

Thank you for the suggestion. Our neural network (called DeepHL-Net according to the comment by Reviewer 1) can process any time-series data. However, other functions of DeepHL are tailored to the trajectory data. We added a discussion about this.

(Lines 418–420)

Although many functions of DeepHL are tailored to a trajectory analysis, DeepHL-Net can process any type of time-series data. As a part of a future study, we plan to apply our network to other time-series such as sounds emitted by animals.

[Comment 4]

It seems that environmental data can be included along the position. But it's not clear how? Maybe an example on this type of application can be beneficial. If no data is present, why not simulate a dataset, with for example different behavior between A and B when the temperature is changing?

[Answer]

Thank you for the suggestion. Other time-series data such as environmental and biological data can be input into our neural network with the primitive features (speed and relative angular speed). We have added a description about it. In the cricket analysis, we input the rotational speed of the body-axis measured by a treadmill with the primitive features as described in the supplementary information. However, as suggested by the reviewer, this can be a good example of inputting other time-series. Therefore, we added descriptions about this in the main text.

In addition, in the analysis of the seabirds, the absolute coordinates (longitude and latitude) are also fed into the neural network because the absolute coordinates of specific locations such as colonies and feeding sites can affect the behavior of the seabirds, which can be considered environmental information. The longitude values were highly correlated with the attention values for the female seabirds. Because the coastline runs north–south, the distance between the coastline and position is related to the longitude of the position. Therefore, this fact indicates that the behavior of the female seabirds is strongly affected by the distance from the coastline. We also added descriptions about this in the main text.

(Lines 464–467)

Normally, the inputs of DeepHL-Net are two-dimensional time-series, i.e., speed and relative angular speed.

When we input an additional time-series (such as the original coordinates) into DeepHL-Net, the additional time-series are added as additional dimensions of the inputs.

(Lines 311–318)

In this analysis, in addition to the speed and relative angular speed, we input additional sensor data measured using a treadmill, i.e., a rotational speed of the body-axis computed from a body-axis angle measured using the treadmill, into DeepHL-Net. Fig. 6a shows typical trajectories colored by the attention values of a discriminator layer. DeepHL shows that the rotational speed of the body-axis transiently elevated and peaked earlier in the pre-stimulated group (Supplementary Information, *Application to the study of crickets*; Fig. 6b), indicating that the sound preceding the air puff provoked the prompt rotational changes of the body-axis.

(Lines 323–332)

In this analysis, in addition to the speed and relative angular speed, we input the absolute coordinates (longitude and latitude) into DeepHL-Net because the absolute coordinates of the specific places such as colonies and feeding sites can affect the behavior of the seabirds. The longitude values were highly correlated with the attention values for the female seabirds. Because the coastline runs north--south, the distance between the coastline and a position is related to the longitude of the position. Therefore, this fact indicates that the behavior of the female seabirds is strongly affected by the distance from the coastline (see Supplementary Information, *Application to the study of seabirds*). As described above, because we can input an additional time-series in addition to the speed and relative angular speed into DeepHL, we can see the effect of the time-series on the animal behaviors.

[Comment 5]

This work can be useful to many fields, and maybe the authors could give more examples, which will help the diffusion of their work. I personally think of livestock farming. For example social network for domesticated animals, i.e. to determine hierarchy of the individual within a flock. Maybe in animal health and welfare, can we distinguish between diseased and non-diseased animals?

[Answer]

Thank you for your suggestion. We added the following description about the potential applications of our method.

(Lines 423–425)

In addition, livestock farming has many potential applications of our animal behavior analysis. For example, our method can be applied to identifying characteristic behaviors of disease animals, productive cows, and

submissive cows in a social hierarchy.

[Comment 6]

Paragraph starting line 115. You could maybe mentioned the color of the layers in Figure 2 when you introduce them.

[Answer]

Thank you for the comment. We modified the description by describing the layers based on their colors, as shown in Fig. 2f.

[Comment 7]

Line 129. I guess it's not exactly equation 1 because of the softmax function. Softmax is not in the neuron definition?

[Answer]

Thank you for the comment. "Softmax" in Equation (1) is a function (activation function of a neuron). This type of equation is usually used in machine learning studies. To help understand the meaning of Equation (1), we added definitions of the softmax and tanh functions.

(Please refer to line 488)

[Comment 8]

Line 418, I wonder how well the deep learning model is, compared to other classical classifiers (e.g. SVM, random forest etc)? Same Line 509.

[Answer]

Thank you for the comment. The Supplementary Information section (*Comparison with classic approaches*) provides a comparison with a classical approach (decision tree, which provides interpretable rules). In addition to the classification accuracies, we introduced a rule obtained by the classic approach for each animal. However, we could not discover new findings from the rules. We added a discussion about this in the revised version of the manuscript.

(Lines 373–377)

Whereas the classification accuracy of a classic approach is not extremely different from that of DeepHL, as shown in Supplementary Information, *Comparison with classic approaches*, it was difficult to find

fine-grained characteristics of animal behaviors by using the classic approach because it employs only high-level features prepared in advance extracted from a whole trajectory.

[Comment 9]

Line 428, 0.5 dropout. Fixed manually? Based on some tests? Can it be tuned?

[Answer]

Thank you for the comment. In DeepHL, 0.5 is the default value of the dropout rate, and this value was used in our analysis of the 6 animals. Because we wanted to use the same experimental parameters throughout the analysis, we used 0.5, which is used in the original dropout study (Srivastava et al., 2014). However, a user can change the value when using our Python-based software. We added a description about the setting of the dropout rate as follows:

(3rd paragraph in p. 7, Supplementary Information)

The user can configure several parameters such as the dropout rate and network size. For more detail, refer to the readme document associated with the Python files.

REVIEWERS' COMMENTS

Reviewer #1 (Remarks to the Author):

The authors have taken into serious consideration all of the comments and suggestions. Now the level of detail is more appropriate for reproduction and the better report of results seems adequate. The evidence provided convinces me that it can be a useful method.

I am happy for my name to appear together with the review, so according to the rules of the journal I need to sign here.

SIGNATURE: Gonzalo G. de Polavieja and Francisco J. H. Heras.

Reviewer #2 (Remarks to the Author):

Thanks to the authors for taking into account all the comments and change the manuscript accordingly. I think the article is ready for publication and I have no other comments.

Congratulations to the authors for their very interesting work.